# Relational Conformal Prediction for Correlated Time Series

Andrea Cini [1 2]  Alexander Jenkins [1 3]  Danilo Mandic [3]  Cesare Alippi [1 4]  Filippo Maria Bianchi [5 6]

## Abstract

We address the problem of uncertainty quantification in time series forecasting by exploiting observations at correlated sequences. Relational deep learning methods leveraging graph representations are among the most effective tools for obtaining point estimates from spatiotemporal data and correlated time series. However, the problem of exploiting relational structures to estimate the uncertainty of such predictions has been largely overlooked in the same context. To this end, we propose a novel distribution-free approach based on the conformal prediction framework and quantile regression. Despite the recent applications of conformal prediction to sequential data, existing methods operate independently on each target time series and do not account for relationships among them when constructing the prediction interval. We fill this void by introducing a novel conformal prediction method based on graph deep learning operators. Our approach, named *Conformal Relational Prediction* (COREL), does not require the relational structure (graph) to be known a priori and can be applied on top of any pre-trained predictor. Additionally, COREL includes an adaptive component to handle non-exchangeable data and changes in the input time series. Our approach provides accurate coverage and achieves state-of-the-art uncertainty quantification in relevant benchmarks.

## 1. Introduction

Many recent advancements in deep learning methods for time series forecasting rely on learning from large collections of (related) time series (Benidis et al., 2022; Liang et al., 2024). In many application domains, such time series are characterized by a rich spatiotemporal dependency structure that can be exploited by introducing inductive biases in the forecasting architecture (Cini et al., 2023a), to steer the learning procedure toward the most plausible models. Accounting for the existing dependencies, usually represented as a graph, allows the resulting models to obtain accurate predictions with a reduced sample complexity (Jin et al., 2023; Cini et al., 2023a). Besides the accuracy of the point estimates, the *reliability* of the forecasts is a critical aspect of the problem and a key element to enable effective decision-making in many applications (Makridakis, 1996; Petropoulos et al., 2022). Uncertainty quantification methods (Smith, 2024; Vovk et al., 2005) can improve reliability by providing confidence intervals on the forecasting error magnitude, allowing for making more informed decisions (Hyndman and Athanasopoulos, 2018). This is particularly true for risk-sensitive applications such as healthcare (Makridakis et al., 2019) and load forecasting (Gasparin et al., 2022). In this context, inter-series (spatiotemporal) dynamics offer both a challenge and an opportunity. Indeed, while these dependencies can lead to wide prediction intervals (PIs) if overlooked, they may also provide additional knowledge to reduce uncertainty (Zambon and Alippi, 2022).

Existing probabilistic forecasting frameworks often rely on strong distributional assumptions and major modifications of the base point predictor (Benidis et al., 2022; Salinas et al., 2020). As such, they cannot be used to quantify uncertainty given a pre-trained forecasting model. In such a setting, *conformal prediction* (CP) (Vovk et al., 2005; Angelopoulos et al., 2023) methods are particularly appealing. CP is an uncertainty quantification framework that estimates confidence intervals with marginal coverage guarantees from observed prediction residuals. One of the main assumptions of standard CP methods is that of exchangeability between the data used to estimate the confidence intervals and the test data points, i.e., the assumption that their joint probability distribution is invariant to the ordering of the associated sequence of random variables (Angelopoulos et al., 2024). Although this assumption does not usually hold when operating on time series (Barber et al., 2023), several methods have successfully adapted CP to estimate forecast uncertainty (Stankeviciute et al., 2021; Xu and Xie, 2023a;b; Jensen et al., 2022; Auer et al., 2023). Nevertheless, existing CP approaches operate on each (possibly

[1]IDSIA USI-SUPSI, Università della Svizzera italiana [2]Swiss National Science Foundation Postdoc Fellow [3]Imperial College London [4]Politecnico di Milano [5]UiT The Arctic University of Norway [6]NORCE Norwegian Research Centre AS. Correspondence to: Andrea Cini <andrea.cini@usi.ch>.

*Proceedings of the 42nd International Conference on Machine Learning*, Vancouver, Canada. PMLR 267, 2025. Copyright 2025 by the author(s).

multivariate) time series independently and cannot account for dependencies among correlated time series.

In this paper, we propose *Conformal Relational Prediction* (COREL), a novel CP approach leveraging graph representations and graph deep learning (GDL) for quantifying uncertainty in correlated time series forecasting. In our framework, a spatiotemporal graph neural network (STGNN) (Jin et al., 2023; Cini et al., 2023a) is trained on a calibration set to approximate the quantile function of the distribution of prediction residuals. Relationships among time series, assumed to be *sparse*, are learned end-to-end from the observed residuals owing to a graph structure learning module integrated into the processing. Our approach estimates the error quantile function for each time series at each time step, by conditioning the shared uncertainty quantification model on past observations at neighboring nodes (as defined by the learned graph structure). Finally, an adaptive component is added to handle potential non-stationarities by relying on a small set of parameters specific to each time series. Our approach can be applied to the residuals generated by *any* point forecasting model, even those that completely disregard potential relationships among the input time series.

Our main novel contributions can be summarized as follows.

- The first application of GDL to CP for time series;

- A novel, sound, and effective CP method able to quantify uncertainty from observations across a collection of correlated time series;

- A family of graph-based architectures to estimate uncertainty that shares most of the learnable parameters among the processed time series, while including node-level parameters that dynamically adapt to changes in each target sequence.

Empirical results show that COREL achieves state-of-the-art performance compared to existing CP approaches for time series in several datasets and under different scenarios.

## 2. Preliminaries

This section introduces the problem settings and the preliminary concepts that serve as foundations for our approach.

### 2.1. Problem Formulation

Consider a collection of $N$ correlated time series. Denote by $x_t^i \in \mathbb{R}$ the scalar target variable associated with the $i$-th time series at time step $t$; $X_t \in \mathbb{R}^{N \times 1}$ indicates the $N$ stacked target variables w.r.t. the entire time series collection. $X_{t:t+T}$ indicates the sequence within time interval $[t, t+T)$; conversely, with the shorthand $X_{<t}$ refers to

observations up to time step $t$ (excluded). Time series are assumed to be *homogenous*, i.e., all the variables (observables) describe the same physical quantity (e.g., temperature or energy consumption). Analogously, $U_t \in \mathbb{R}^{N \times d_u}$ indicates the $d_u$-dimensional exogenous covariates associated with each time series. We assume that the $i$-th time series is generated by a stochastic *time-invariant* process such as

$$x_t^i \sim p\left(x_t^i | X_{<t}, U_{<t}\right). \qquad (1)$$

Let us hypothesize the existence of a *sparse* predictive causality *à la Granger* (Granger, 1969), i.e., we assume that the values of a single time series are related to the values of a (*small*) subset of other time series in the collection. The extension of the framework to collections of multivariate time series is orthogonal to the proposed approach (e.g., see Feldman et al. 2023); we focus on the univariate case to maintain a contained scope. The problem of dealing with non-stationary processes will be discussed in Sec. 3.4.

**Forecasting** We are interested in a model that produces point forecasts by predicting the unknown $H$-steps-ahead ($H \geq 0$) observation $X_{t+H}$ given a window $W \geq 1$ of past observations $X_{t-W:t}$ and the associated exogenous variables $U_{t-W:t}$ as

$$\widehat{X}_{t+H} = \mathcal{F}_{\theta}(X_{t-W:t}, U_{t-W:t}). \qquad (2)$$

$\mathcal{F}_{\theta}$ denotes a generic parametric model family, i.e., a simple recurrent neural network (RNN) for univariate time series. Given a trained model, our objective is to build a confidence interval around predictions $\widehat{X}_{t+H}$. Note that the following easily extends to multi-step predictions $\widehat{X}_{t:t+H}$, but we focus on forecasting the single time step $H$ to simplify the presentation and discussion.

**Uncertainty quantification** Our objective is to estimate PIs, $C_{i,t}^{\alpha}(\widehat{X}_{t+H})$, such that

$$P\left(x_{t+H}^i \in C_{i,t}^{\alpha}\left(\widehat{X}_{t+H}\right)\right) \geq 1 - \alpha, \qquad (3)$$

where $\alpha$ is the desired confidence level. If the interval satisfies Eq. 3, we say that the PI achieves marginal coverage $1 - \alpha$. Similarly, we say that the PI provides conditional coverage $1 - \alpha$ if

$$P\left(x_{t+H}^i \in C_{i,t}^{\alpha}\left(\widehat{X}_{t+H}\right) \Big| X_{<t}, U_{<t}\right) \geq 1 - \alpha. \qquad (4)$$

Conditional coverage provides stronger guarantees and it is often harder to achieve (Angelopoulos et al., 2024). In the following, we will omit the dependence of the interval on the forecasts and simply write $C_{i,t}^{\alpha}$. Among uncertainty quantification methods, we are interested in post-hoc approaches that can build confidence intervals for any given pre-trained point predictor $\mathcal{F}_{\theta}$ without requiring any modification of the base forecasting architecture.

## 2.2. Conformal Prediction

As anticipated in Sec. 1, standard CP methods (Vovk et al., 2005; Angelopoulos et al., 2023) are a class of distribution-free uncertainty quantification techniques that build PIs from empirical quantiles of *conformal scores*. In the forecasting setting, we consider as conformal scores the prediction residuals,

$$\boldsymbol{r}_t^i = \boldsymbol{x}_t^i - \hat{\boldsymbol{x}}_t^i, \tag{5}$$

and use $\boldsymbol{R}_t$ to denote residuals w.r.t. the entire time series collection. Under appropriate assumptions, CP methods can build valid and informative PIs (Angelopoulos et al., 2023; Barber et al., 2023). Split conformal prediction (SCP) (Vovk et al., 2005) is arguably the most common approach and exploits scores computed on a *calibration set* that is disjoint from the training data (i.e., a post-hoc approach).

As mentioned, most standard CP methods rely on the assumption that calibration and test data are *exchangeable*, which allows the procedure to treat them symmetrically and obtain valid (marginal) coverage guarantees (Angelopoulos et al., 2024). Since this assumption does not hold when dealing with time series data, there have been several recent results extending the CP framework beyond exchangeability (Tibshirani et al., 2019; Stankeviciute et al., 2021; Gibbs and Candes, 2021; Xu and Xie, 2023a). In particular, Barber et al. (2023) showed that approximate coverage can be achieved by reweighting the residuals to account for the lack of exchangeability between calibration and test set. Auer et al. (2023) learn such a reweighting scheme through an attention-based architecture. Differently, Xu and Xie (2023b) introduce SPCI, a method based on fitting a quantile random forest (Meinshausen and Ridgeway, 2006) on the most recent prediction residuals at each time step. Similar to SPCI, our approach relies on quantile regression to build PIs but differently from existing methods, it exploits observations in arbitrary sets of time series by relying on GDL operators.

## 2.3. Quantile Regression

Quantile regression (Koenker and Hallock, 2001) is an established statistical framework that consists of learning a model of the quantile function (the inverse c.d.f.) of a target distribution from observations. In particular, given $\boldsymbol{y} \sim p(\boldsymbol{y}|\boldsymbol{x})$ and observations $(x_1, y_1), \ldots, (x_N, y_N)$, a standard approach to estimate the $\alpha$-quantile is to train a model by minimizing the so-called pinball loss

$$\ell^\alpha(\hat{q}^\alpha(x), y) = \begin{cases} (1-\alpha)(\hat{q}^\alpha(x) - y), & \hat{q}^\alpha(x) \geq y \\ \alpha(y - \hat{q}^\alpha(x)), & \hat{q}^\alpha(x) < y \end{cases} \tag{6}$$

where $\hat{q}^\alpha(x)$ is the estimate of the $\alpha$-quantile w.r.t. $x$.

**Quantile networks** Quantile regression has been incorporated in several probabilistic forecasting architectures (Beni-dis et al., 2022). The simplest approach consists of using a multi-output network to predict a set of quantiles of interest and interpolate among them to approximate the entire quantile function (Wen et al., 2017). More complex approaches rely on, e.g., splines (Gasthaus et al., 2019). Conversely, implicit quantile networks (IQNs) (Dabney et al., 2018; Ostrovski et al., 2018; Gouttes et al., 2021) approximate the quantile function by being trained to minimize the loss in Eq. 6 given the quantile level $\alpha$ as input and sampling a random $\alpha$ for each sample in a mini-batch.

## 2.4. Graph Deep Learning for Time Series Forecasting

Graph neural networks (GNNs) (Bacciu et al., 2020; Bronstein et al., 2021) process graph-structured data by incorporating the graph topology as an inductive bias, e.g., by relying on message-passing layers (Gilmer et al., 2017). STGNNs (Jin et al., 2023; Cini et al., 2023a) leverage message-passing layers within sequence modeling architectures to process spatiotemporal data and collections of time series where dependencies are represented as a (possibly dynamic) graph. We consider as reference architectures time-then-space (TTS) models (Gao and Ribeiro, 2022; Cini et al., 2023a) where each time series in the collection is processed independently from the others by a temporal encoder whose output is then fed into a stack of GNN layers. In particular, we adopt the following template architecture:

$$\boldsymbol{h}_t^{i,0} = \text{SEQENC}\left(\boldsymbol{x}_{t-W:t}^i, \boldsymbol{u}_{t-W:t}^i\right), \tag{7}$$

$$\boldsymbol{H}_t^{l+1} = \text{GNN}_l(\boldsymbol{H}_t^l, \boldsymbol{A}),, \quad l = 0, \ldots, L-1 \tag{8}$$

$$\hat{\boldsymbol{y}}_t^i = \text{READOUT}\left(\boldsymbol{h}_t^{i,L}\right), \tag{9}$$

where $\boldsymbol{A} \in \mathbb{R}^{N \times N}$ is the graph adjacency matrix and $\hat{\boldsymbol{y}}_t^i$ a generic node-level prediction associated with the problem at hand. $\text{SEQENC}(\cdot)$ and $\text{GNN}_l(\cdot)$ denote, respectively, any sequence modeling architecture, e.g., an RNN, and any GNN layer, e.g., based on message-passing. Representations can then be mapped into predictions $\widehat{\boldsymbol{Y}}_t$ by using any readout block, e.g., a multilayer perceptron (MLP). STGNNs have been used as forecasting architecture ($\boldsymbol{Y}_t = \boldsymbol{X}_{t+H}$) with great success. In the following, we will exploit this framework as a backbone for estimating the residual quantile distribution. We refer to Sec. 4 and Jin et al. (2023) for more discussion on the application of STGNNs in the context of time series analysis.

## 3. Conformal Relational Prediction

Our objective is to build PIs by exploiting relational dependencies across the residuals of the target time series. We model the dependencies as edges of a graph and learn them under the assumption that the relational structure is *sparse*, which reduces the computational costs and act as an inductive bias on the structure learning architecture. By relying on

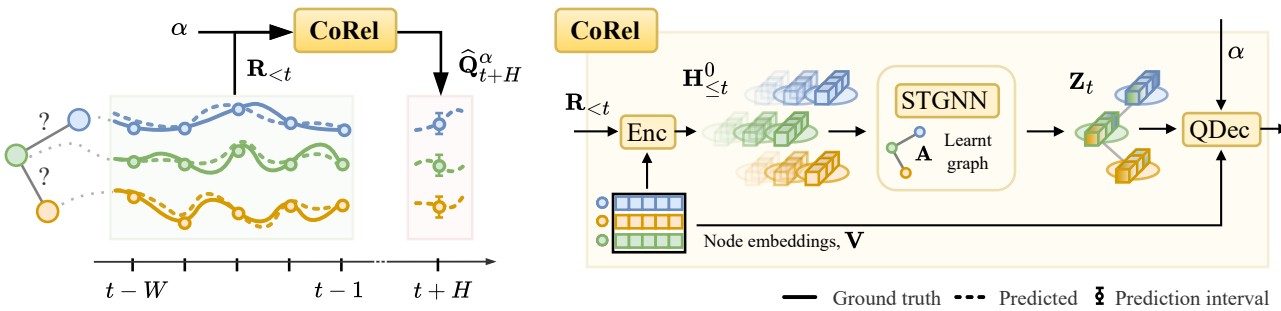

Figure 1: Overview of CoREL. Past residuals are used as input to a hybrid global-local graph-based quantile network.

such representation, we can leverage GDL methods for time series to process the data. In particular, we train a STGNN on the residuals of the calibration set to predict the quantiles of the error distribution. Conditioning the prediction on the recent history of related time series allows for taking the dependency structure of the data into account when estimating uncertainty: a key aspect in applying conformal prediction to non-exchangeable data (Barber et al., 2023). Compared to existing methods (Xu and Xie, 2023b; Auer et al., 2023) that only capture temporal dependencies, our approach allows for modeling spatiotemporal dependencies among different time series. Sec. 3.1 presents the details of the proposed conformal inference procedure by assuming that the relational structure at each time step is defined by an adjacency matrix $A \in \mathbb{R}^{N \times N}$ (Sec. 3.1). We then show how to learn the graph structure directly from data and make the model adaptive in Sec. 3.2 and Sec. 3.4, respectively. Finally, we discuss the theoretical properties of the approach in Sec. 3.3. Fig. 1 shows an overview of the architecture.

### 3.1. Relational Quantile Predictor

Consider a standard SCP setup, where the training data are split into training and calibration sets. For the moment, we disregard possible nonstationarities in the data considering the problem setup introduced in Sec. 2.1 and encode spatial dependencies in the adjacency matrix $A \in \mathbb{R}^{N \times N}$. While the training set is used to fit the point predictor $\mathcal{F}_\theta$, we use the prediction residuals in the calibration set ($\mathcal{R}^{cal}$) to learn the quantile function of the error distribution at each step.

**Relational quantile regression**   We implement the quantile regressor as a hybrid global-local STGNN, which mixes global (shared) parameters with local, target-specific components (Smyl, 2020). Sharing most learnable parameters across all time series reduces sample complexity, while local parameters allow for tailoring the processing to each series. Specifically, we keep all processing blocks shared and associate a learnable node embedding $v^i \in \mathbb{R}^{d_v}$ with each time series (Cini et al., 2023b). More specifically, our model is a quantile network (see Sec. 2.3) composed of the

following processing layers:

$$h_t^{i,0} = \text{ENC}\left(r_{t-1}^i, v^i\right), \qquad (10)$$

$$Z_t = \text{STGNN}\left(H_{\leq t}^0, A\right), \qquad (11)$$

$$\hat{q}_{t+H}^{i,\alpha} = \text{QDEC}\left(\alpha, z_t^i, v^i\right), \qquad (12)$$

where $r_{t-1}^i$ are prediction residuals (Eq. 5) and $\hat{q}_{t+H}^{\alpha,i}$ is the predicted $\alpha$-quantile at time step $t + H$ for the $i$-th time series. ENC($\cdot$) denotes any encoding layer, e.g., a linear transformation or an MLP. For the STGNN block, several designs are possible (e.g., see Jin et al. 2023); the one we follow is the template in Sec. 2.4. QDEC($\cdot$) is a readout mapping the representations at each node to the prediction of the quantile of specified level $\alpha$. We refer to the family of quantile networks defined in Eq. 10–12 as *relational quantile predictor*s (RelQPs) and use the notation

$$\widehat{Q}_t^\alpha = \mathcal{Q}_\psi\left(\alpha, V; R_{t-W:t}, A\right), \qquad (13)$$

where $\mathcal{Q}_\psi$ indicates the shared (global) part of the network and $\widehat{Q}_t^\alpha \in \mathbb{R}^N$ denotes the predicted $\alpha$-quantiles at time step $t$ w.r.t. the full time series collection. Note that the framework can easily accommodate further inputs at the encoding block (e.g., we can condition the regression on $X_{<t}$ and $U_{<t}$). The model is trained by minimizing the pinball loss (Eq. 6) at each time step in the calibration set w.r.t. the full-time series collection. Through the message-passing layers, the residuals of each time series contribute to estimating the quantiles of the error distribution at neighboring nodes. In practice, we restrict the input of the regressor to the most recent observations rather than considering the full sequence (the window length here can also be different from the one used by the point predictor).

**Building the confidence intervals**   Given the trained quantile network $\mathcal{Q}_\psi$, we build the PIs for each target (test) time step as

$$\widehat{C}_{i,t}^\alpha = \left[\hat{x}_{t+H}^i + \hat{q}_t^{i,\alpha/2}, \, \hat{x}_{t+H}^i + \hat{q}_t^{i,1-\alpha/2}\right], \qquad (14)$$

or

$$\hat{\beta}_i = \arg\min_{\beta_i} \left| \hat{q}_t^{i,1-\alpha/2+\beta_i} - \hat{q}_t^{i,\alpha/2+\beta} \right| \tag{15}$$

$$\widehat{C}_{i,t}^{\alpha} = \left[ \hat{x}_{t+H}^i + \hat{q}_t^{i,\alpha/2+\hat{\beta}_i}, \hat{x}_{t+H}^i + \hat{q}_t^{i,1-\alpha/2+\hat{\beta}_i} \right], \tag{16}$$

where $\widehat{C}_{i,t}^{\alpha}$ indicates the estimated PI. While both Eq. 14 and Eq. 16 correspond to the same confidence level, the PI in Eq. 16 can be narrower, at the expense of the additional computation needed to obtain $\hat{\beta}$ (Xu and Xie, 2023a). In practice, one can choose between the two approaches given computational constraints and the difference in performance observed on a validation set. Note that $\widehat{C}_{i,t}^{\alpha}$ can provide only *approximate* coverage, as it is subject to approximation errors of the true quantile function. Sec. 3.3 will discuss this aspect in detail. A potential drawback of COREL is that residuals cannot be assumed exchangeable in most practical scenarios. The error distribution can be *non-stationary*, making it difficult to obtain any coverage guarantee. To mitigate the problem, Sec. 3.4 discusses an efficient and scalable approach to make the framework adaptive by updating local components of the architecture over time. Finally, it is worth noting that the use of relational components in COREL relies on the actual presence of the associated dependencies in the data. In practical applications, the presence of spatiotemporal correlations in the residuals can be verified through ad-hoc statistical tests (Zambon and Alippi, 2022; 2023), whose outcome can support the adoption of COREL.

### 3.2. Learning the Relational Structure

Assuming the dependency structure across time series to be unknown, we integrate a graph learning module into the architecture to derive the operational graph topology directly from the residuals. To do so, we adopt a probabilistic structure learning framework (Niculae et al., 2023; Cini et al., 2023c; Manenti et al., 2024). In particular, we associate each edge with a score $\phi^{ij}$ and learn a distribution over $K$-NN graphs parametrized by the matrix $\boldsymbol{\Phi} \in \mathbb{R}^{N \times N}$ (Cini et al., 2023c; Kazi et al., 2022). Notably, we consider graphs obtained by sampling, for each $i$-th node, $K$ elements *without replacement* from the categorical distribution

$$\boldsymbol{\Phi} = \mathcal{E}_{\boldsymbol{\xi}}\left(\boldsymbol{R}_{<t}, \boldsymbol{V}, \dots\right) \tag{17}$$

$$M_i = \text{Categorical}\left(\frac{\exp\{\phi^{ik}\}}{\sum_{j=1}^N \exp\{\phi^{ij}\}}; k \in \{1, \dots, N\}\right), \tag{18}$$

where $\mathcal{E}_{\boldsymbol{\xi}}(\cdot)$ is a generic trainable encoder with parameters $\boldsymbol{\xi}$. In practice, sampling can be done efficiently by exploiting the GumbelTopK trick (Kool et al., 2019) and scores $\boldsymbol{\Phi}$ can be parametrized directly as $\boldsymbol{\Phi} = \boldsymbol{\xi}$.

**End-to-end learning** To propagate gradients through the sampling, we rely on the continuous relations introduced by

Xie and Ermon (2019) paired with a straight-through gradient estimator (Bengio et al., 2013) to obtain discrete samples. Optionally, we sparsify the gradients by backpropagating only through a random subset of the zero entries of $\boldsymbol{A}$ (more details are provided in App. D). As already mentioned, different parametrizations and gradient estimators for subset samplers exist and can be considered. Furthermore, if the sparsity assumption is deemed unrealistic for the problem at hand, other distributions (e.g., based on Bernoulli random variables) can be considered (Cini et al., 2023c).

### 3.3. Theoretical Analysis and Further Discussion

We start the discussion by providing an intuitive bound on the approximate coverage provided by COREL.

**Proposition 3.1.** *Let* $P_{t+H}^c(\boldsymbol{x}_{t+H}^i) = p_{t+H}(\boldsymbol{x}_{t+H}^i \mid \boldsymbol{X}_{<t}, \boldsymbol{U}_{<t})$ *and* $P_{\psi}^c(\boldsymbol{x}_{t+H}^i) = p_{\psi}(\boldsymbol{x}_{t+H} \mid \boldsymbol{X}_{<t}, \boldsymbol{U}_{<t})$ *be the true conditional data-generating distribution at the test point* $t + H$ *and the probability distribution associated with the learned quantile function* $\mathcal{Q}_{\psi}$, *respectively. Then*

$$P_{t+H}^c\left(\boldsymbol{x}_{t+H}^i \in \widehat{C}_{i,t}^{\alpha}(\widehat{\boldsymbol{X}}_{t+h})\right) \geq 1 - \alpha - TV\left(P_{\psi}^c, P_{t+H}^c\right)$$

*where* $TV(\,\cdot\,)$ *denotes the total variation function.*

The proof relies on the properties of the total variation of probability measures and can be found in App. A. Here, differently from the problem settings introduced in Sec. 2.1, we do not assume the process to be time-invariant. Prop. 3.1 links the *conditional* coverage gap to the approximation error in estimating the quantile function of the residuals. The bound provided in Prop. 3.1 shares similarities with the one in (Barber et al., 2023), which bounds the miscoverage gap for CP from weighted empirical quantiles. Prop. 3.1 can be seen as an analogous result that holds when estimates obtained from empirical quantiles are replaced with a parametric function approximation. By making assumptions on the expressivity of the quantile regressor in Eq. 13 and on the stationarity process (e.g., by assuming a strongly mixing process), we can expect the total variation between the learned and true distribution to shrink asymptotically as the size of the calibration set increases. Moreover, in this case, monitoring the coverage gap on a validation set offers an estimate of the actual miscoverage on test data. Similar analyses have been carried out for recently introduced CP methods for time series (Xu and Xie, 2023b; Lee et al., 2025); we refer the reader to these related works. If we instead expect the process to be non-stationary, $\mathcal{Q}_{\psi}$ has to be updated over time to keep the coverage gap contained. Within this context, the next section discusses a simple and sample-efficient approach to make COREL adaptive. Finally, the computational complexity of COREL will depend on the STGNN used to implement the RelQP and the number of edges sampled while learning the relational structure.

For example, the cost of a forward pass for a standard TTS STGNN with window size $W$ and a graph with $E$ edges would scale as $\mathcal{O}(WN + E)$ (Cini et al., 2023a).

### 3.4. Adaptation

The RelQP model introduced in Sec. 3.1 can yield arbitrarily large coverage gaps in the presence of distribution shifts from the calibration set where the model is trained. Adopting a re-training approach, such as in Xu and Xie (2023b), would be impractical due to the higher sample complexity entailed by the deep learning approach that we adopt. Therefore, to mitigate this issue while keeping the computational complexity under control, we update only the local components of the model over time, i.e., the learnable node embeddings $V$ (Cini et al., 2023b). This allows for keeping most of the learnable parameters fixed and fine-tuning only a small number of weights for each node. Empirically, we show that this procedure can effectively improve the quality of the uncertainty estimates.

## 4. Related Work

The problem of quantifying forecast uncertainty is central in fundamental and applied research in time series forecasting (Hyndman and Athanasopoulos, 2018; Petropoulos et al., 2022). Among deep learning approaches (Benidis et al., 2022), many generative architectures have been proposed as means to obtain probabilistic forecasts (Salinas et al., 2020; Rangapuram et al., 2018; de Bézenac et al., 2020; Rasul et al., 2021). Most related to our approach are those methods that exploit quantile regression (Wen et al., 2017; Gasthaus et al., 2019; Kan et al., 2022; Gouttes et al., 2021). Similarly to COREL, these quantile regression techniques do not usually require strong assumptions on the data distribution.

**Uncertainty quantification in STGNNs**  Regarding probabilistic graph-based forecasting architecture, the existing literature is limited (Jin et al., 2023; Cini et al., 2023a). Wu et al. (2021) investigate the combination of STGNNs with standard uncertainty quantification techniques for deep learning. Pal et al. (2021) use an STGNN to implement a state-space model and quantify uncertainty within a Bayesian framework. Wen et al. (2023) propose a probabilistic predictor based on combining STGNNs with a diffusion model (Ho et al., 2020). Zambon et al. (2023) introduce a framework for designing probabilistic graph state-space models that can process collections of time series. However, all these methods cannot operate on top of an existing pre-trained model and require training an ad-hoc forecasting model. Conversely, COREL is trained, within a CP framework, on predicting the quantiles of the error distribution of any existing model, rather than on forecasting the target variable.

**Conformal prediction**  Related work on CP for time series has been already discussed in Sec. 2.2 and Sec. 3. Related to our method, Mao et al. (2024) propose a CP approach for (static) spatially correlated data. Jiang et al. (2024) propose to quantify the uncertainty in predicting power outages by fitting a quantile random forest (Meinshausen and Ridgeway, 2006) on time series from neighboring geographical units. COREL can be framed among the CP methods that learn a model of conformal scores distribution (Xu and Xie, 2023b; Lee et al., 2024). Differently from existing methods that operate on each time series separately, the estimates are conditioned on errors at both the target time series as well as at neighboring nodes. To the best of our knowledge, no previous CP method has been designed to specifically operate on collections of correlated time series and exploit graph deep learning operators. CP methods for multivariate time series do exist (Xu et al., 2024; Sun and Yu, 2024; Feldman et al., 2023), but operate on a single multidimensional time series. Moreover, although global-local models are popular among forecasting architectures (Smyl, 2020; Benidis et al., 2022), COREL is the first CP architecture of this kind. Finally, CP methods have also been applied to *static* graphs and used to quantify the uncertainty of GNNs, both in inductive (Zargarbashi et al., 2023) and transductive (Huang et al., 2024) settings. These methods often assume node/edge exchangeability (Zargarbashi et al., 2023; Huang et al., 2024) or are limited to node classification (Clarkson, 2023) or link prediction (Zhao et al., 2024). Recently, Davis et al. (2024) proposed a CP method for node classification with GNNs in dynamic networks.

## 5. Experiments

We validate COREL across three experimental settings. In the first one (Sec. 5.1), we compare it against state-of-the-art CP methods operating on the residuals produced by different forecasting models. Then, we analyze COREL in a controlled environment (synthetic dataset). Finally, we assess the effectiveness of the procedure described in Sec. 3.4 in adaptively improving the PIs. We implement **COREL** as an RNN followed by two message-passing layers. To approximate the quantile function, we train the model by minimizing the pinball loss over a discrete set of quantiles, similarly to Wen et al. (2017). PIs are constructed as in Eq. 14; App. G.2 shows results for the alternative construction in Eq. 16. To learn the graph, we directly parametrize the score matrix $\Phi$ by associating a learnable parameter with each of its entries. We use as metrics: 1) the difference between the specified confidence level $1 - \alpha$ and the observed coverage on the test set ($\Delta Cov$), 2) the width of the PI (*PI-Width*), and 3) the Winkler score (Winkler, 1972), which is computed as the width of the PI plus penalty for each observation outside of the predicted interval proportional to the actual error (*Winkler*). Note that balancing

| | | Metric | SCP | NexCP | SeqCP | SPCI | HopCPT | CoRNN | CoREL |
|---|---|---|---|---|---|---|---|---|---|
| METR-LA | RNN | ΔCov | -1.28 | -1.00 | -6.93 | $-1.24_{\pm0.01}$ | $-1.46_{\pm0.11}$ | $-0.49_{\pm0.53}$ | $-1.29_{\pm0.21}$ |
| | | PI-Width | 20.69 | 25.23 | 19.52 | $19.84_{\pm0.01}$ | $16.62_{\pm0.10}$ | $19.48_{\pm0.43}$ | $14.38_{\pm0.22}$ |
| | | Winkler | 40.80 | 41.33 | 50.12 | $37.95_{\pm0.01}$ | $25.63_{\pm0.19}$ | $30.24_{\pm0.18}$ | $\mathbf{23.78_{\pm0.20}}$ |
| | TRANSF | ΔCov | -1.18 | -0.97 | -6.98 | $-1.13_{\pm0.00}$ | $-1.22_{\pm0.56}$ | $-0.55_{\pm0.61}$ | $-1.02_{\pm0.63}$ |
| | | PI-Width | 20.81 | 25.30 | 19.44 | $19.86_{\pm0.01}$ | $16.66_{\pm0.13}$ | $19.38_{\pm0.64}$ | $14.17_{\pm0.43}$ |
| | | Winkler | 40.55 | 41.64 | 49.90 | $37.74_{\pm0.01}$ | $25.37_{\pm0.33}$ | $30.33_{\pm0.18}$ | $\mathbf{23.65_{\pm0.19}}$ |
| | STGNN | ΔCov | -0.99 | -0.62 | -13.60 | $-0.87_{\pm0.00}$ | $-0.13_{\pm0.27}$ | $-0.34_{\pm0.49}$ | $-0.92_{\pm0.31}$ |
| | | PI-Width | 17.30 | 22.20 | 12.87 | $16.38_{\pm0.01}$ | $15.75_{\pm0.19}$ | $16.20_{\pm0.24}$ | $14.65_{\pm0.24}$ |
| | | Winkler | 34.94 | 34.49 | 40.36 | $33.66_{\pm0.01}$ | $\mathbf{22.80_{\pm0.28}}$ | $28.74_{\pm0.12}$ | $24.70_{\pm0.16}$ |
| CER-E | RNN | ΔCov | -3.46 | 0.11 | -3.57 | $-3.45_{\pm0.00}$ | $-4.37_{\pm0.32}$ | $-2.24_{\pm0.34}$ | $-3.60_{\pm0.35}$ |
| | | PI-Width | 2.60 | 3.26 | 2.75 | $2.39_{\pm0.00}$ | $1.97_{\pm0.03}$ | $1.96_{\pm0.02}$ | $1.83_{\pm0.03}$ |
| | | Winkler | 5.69 | 5.48 | 5.79 | $5.29_{\pm0.00}$ | $3.87_{\pm0.07}$ | $3.84_{\pm0.01}$ | $\mathbf{3.71_{\pm0.04}}$ |
| | TRANSF | ΔCov | -3.35 | 0.10 | -3.53 | $-3.26_{\pm0.01}$ | $-3.95_{\pm0.60}$ | $-2.04_{\pm0.32}$ | $-3.97_{\pm0.24}$ |
| | | PI-Width | 2.52 | 3.16 | 2.67 | $2.33_{\pm0.00}$ | $2.01_{\pm0.09}$ | $1.94_{\pm0.03}$ | $1.80_{\pm0.02}$ |
| | | Winkler | 5.60 | 5.36 | 5.69 | $5.20_{\pm0.00}$ | $3.88_{\pm0.12}$ | $3.82_{\pm0.02}$ | $\mathbf{3.67_{\pm0.02}}$ |
| | STGNN | ΔCov | -4.30 | 0.08 | -3.83 | $-4.17_{\pm0.01}$ | $-5.06_{\pm0.15}$ | $-2.13_{\pm0.83}$ | $-4.99_{\pm0.61}$ |
| | | PI-Width | 2.28 | 3.00 | 2.42 | $2.09_{\pm0.00}$ | $1.79_{\pm0.01}$ | $1.85_{\pm0.04}$ | $1.77_{\pm0.05}$ |
| | | Winkler | 5.11 | 4.87 | 4.99 | $4.76_{\pm0.00}$ | $\mathbf{3.49_{\pm0.02}}$ | $3.72_{\pm0.02}$ | $3.76_{\pm0.04}$ |
| AQI | RNN | ΔCov | 5.09 | -0.63 | -3.21 | $1.79_{\pm0.01}$ | $0.01_{\pm2.62}$ | $-2.06_{\pm1.52}$ | $-2.78_{\pm0.60}$ |
| | | PI-Width | 118.06 | 82.04 | 74.73 | $103.88_{\pm0.03}$ | $90.35_{\pm13.04}$ | $71.61_{\pm1.82}$ | $68.13_{\pm1.30}$ |
| | | Winkler | 148.61 | 131.18 | 135.59 | $143.10_{\pm0.01}$ | $133.24_{\pm8.31}$ | $113.11_{\pm0.77}$ | $\mathbf{107.67_{\pm0.94}}$ |
| | TRANSF | ΔCov | 5.07 | -0.66 | -3.17 | $1.60_{\pm0.01}$ | $-2.34_{\pm1.19}$ | $-1.93_{\pm0.44}$ | $-1.81_{\pm1.67}$ |
| | | PI-Width | 118.50 | 81.15 | 75.08 | $104.11_{\pm0.01}$ | $80.37_{\pm2.09}$ | $73.25_{\pm0.95}$ | $72.25_{\pm3.16}$ |
| | | Winkler | 150.33 | 132.08 | 137.62 | $145.14_{\pm0.02}$ | $129.85_{\pm3.14}$ | $112.71_{\pm0.44}$ | $\mathbf{108.71_{\pm1.38}}$ |
| | STGNN | ΔCov | 4.48 | -0.32 | -2.94 | $2.64_{\pm0.01}$ | $-0.78_{\pm1.34}$ | $-1.52_{\pm0.51}$ | $-2.00_{\pm1.54}$ |
| | | PI-Width | 111.68 | 80.01 | 71.92 | $99.90_{\pm0.03}$ | $79.39_{\pm3.38}$ | $70.17_{\pm0.80}$ | $68.24_{\pm2.65}$ |
| | | Winkler | 143.39 | 127.98 | 130.86 | $137.09_{\pm0.03}$ | $121.26_{\pm2.48}$ | $109.65_{\pm0.30}$ | $\mathbf{108.45_{\pm1.52}}$ |

Table 1: Performance comparison for $\alpha = 0.1$. ΔCov values are color-coded: green (0-2%), yellow (2-3%), orange (3-4%), red (>4%). The lowest Winkler score for each scenario is shown in bold.

coverage and PI width is the main challenge. More details and results are provided in the appendix.

### 5.1. Time Series Forecasting Benchmarks

We consider the following datasets, each coming from a different application domain: **METR-LA** from the traffic forecasting literature (Li et al., 2018); a collection of air quality measurements from different Chinese cities (**AQI**) (Zheng et al., 2015); a collection of energy consumption profiles acquired from smart meters within the CER smart metering project (**CER-E**) (Commission for Energy Regulation, 2016; Cini et al., 2022). We follow the preprocessing steps of previous works (Li et al., 2018; Wu et al., 2019; Cini et al., 2023b) and adopt 40%/40%/20% splits for training, calibration, and testing, respectively. For each dataset, we first train 3 different baseline models: a simple **RNN** with gated recurrent unit (GRU) cells (Cho et al., 2014), a

decoder-only **Transformer** (Vaswani et al., 2017), and a simple TTS **STGNN** obtained by following the template in Sec. 2.4. The latter uses a pre-defined graph that models the dependencies across the time series. After training, we evaluate each baseline on the calibration set and save the associated residuals, which are then used as input to the different CP methods. More details on the datasets and base models are provided in App. C.

**Baselines** We compared CoREL against the following baselines: 1) **SCP**, the standard split CP; 2) **SeqCP**, where, analogously to Xu and Xie (2023a), we compute empirical quantiles using only the most recent $K$ residuals at each time step; 3) **NexCP** (Barber et al., 2023), which computes empirical quantiles by assigning exponentially decaying weights to past residuals; 4) **SPCI** (Xu and Xie, 2023b), which estimates the residuals' quantile function from the last few steps of each time series with a random forest; 5) **HopCPT** which

reweights past residuals by learning attention scores with a Modern Hopfield Network (Ramsauer et al., 2021). These baselines are representative of the current state-of-the-art in CP for time series forecasting; in particular, SPCI (based on quantile regression) and HopCPT (based on reweighting) are representative of the main recent paradigms. Note that a comparison with non-post-hoc methods would be problematic, as results heavily depend on the base predictors being used. We also include in our comparison a model called **CoRNN**, where we use the same architecture as CoREL but remove message-passing layers and node embeddings. CoRNN is an ablation of the introduced designs. Except for HopCPT, which uses a custom procedure (Auer et al., 2023), model selection is performed on a validation set by optimizing the Winkler score.

**Results** Tab. 1 reports the results across the datasets and the base prediction models. The first observation is that CoREL outperforms the competitors in terms of Winkler score in almost all cases. We observed a few exceptions only when the baseline is itself an STGNN, as it is already expected to take care of modeling spatiotemporal dependencies. However, note that the STGNN base model has access to a pre-defined graph, which is not always available in practical applications. In terms of coverage, CoREL achieves good results, with the exception of some cases in the CER-E dataset. However, we note that our model selection prioritized the Winkler score which emphasizes the width of the prediction bands besides the coverage. CoRNN, the simplified version of our approach, obtains good overall performance and is competitive against the state-of-the-art but, despite providing good coverage, it is outperformed by CoREL in most scenarios in terms of Winkler score. Among the competitors, HopCPT provides competitive results in METR-LA and CER-E but with a larger coverage gap. Except for SeqCP, the other baselines obtain good coverage in most settings at the expense of drastically wider PIs. Regarding computational scalability, note that CoREL shares most of the learnable parameters among time series and that its training can be efficiently parallelized on a GPU. Furthermore, besides learnable node embeddings $V$, CoREL relies only on a short window of the most recent observations as an input at each time step. Conversely, SPCI requires training a different model for each time series, and HopCPT requires computing attention scores w.r.t. the entire calibration set at each time step.

## 5.2. Controlled Environment

We evaluated the behavior of CoREL in a controlled environment by simulating a diffusion process on a graph. In particular, the experiment relied on the GPVAR benchmark introduced by Zambon and Alippi (2022), with a setup analogous to (Cini et al., 2023b). Data were generated recursively

Table 2: Performance on GPVAR.

|  | GPVAR-RNN | | | GPVAR-STGNN | | |
|---|---|---|---|---|---|---|
| Models | $\Delta$Cov | PI-Width | Winkler | $\Delta$Cov | PI-Width | Winkler |
| SCP | -0.02 | 1.67 | 2.14 | -0.02 | 1.32 | 1.66 |
| CoRNN | -0.1$_{\pm 0.2}$ | 1.63$_{\pm 0.01}$ | 2.04$_{\pm 0.00}$ | 0.0$_{\pm 0.0}$ | 1.32$_{\pm 0.00}$ | 1.66$_{\pm 0.00}$ |
| CoREL | 0.0$_{\pm 0.1}$ | 1.33$_{\pm 0.00}$ | 1.67$_{\pm 0.00}$ | 0.1$_{\pm 0.1}$ | 1.32$_{\pm 0.01}$ | 1.66$_{\pm 0.00}$ |
| w/ true $A$ | -0.1$_{\pm 0.1}$ | 1.32$_{\pm 0.00}$ | 1.66$_{\pm 0.00}$ | 0.0$_{\pm 0.2}$ | 1.32$_{\pm 0.01}$ | 1.66$_{\pm 0.00}$ |

from the auto-regressive polynomial graph filter:

$$\boldsymbol{H}_t = \sum_{l=1}^{L} \sum_{q=1}^{Q} \Theta_{q,l} \boldsymbol{A}^{l-1} \boldsymbol{X}_{t-q},$$

$$\boldsymbol{X}_{t+1} = a \odot \tanh(\boldsymbol{H}_t) + b \odot \tanh(\boldsymbol{X}_{t-1}) + \eta_t, \quad (19)$$

where parameters $\boldsymbol{\Theta} \in \mathbb{R}^{Q \times L}$, $a \in \mathbb{R}$, $b \in \mathbb{R}$ are kept fixed across nodes and $\eta_t \sim \mathcal{N}(\boldsymbol{0}, \sigma^2 \mathbb{I})$ with $\sigma = 0.4$. We ran the simulation on a graph with 60 nodes and with a topology analogous to previous works (Cini et al., 2023b); more details are provided in App. C. We use both a RNN and STGNN as base point predictors. As shown by Zambon and Alippi (2022), STGNNs can obtain a forecasting accuracy near the theoretical optimum in this dataset, which results in uncorrelated residuals. As such, we would expect standard SCP to be sufficient when using an STGNN as base model. The objective of this experiment is to show that CoREL is effectively able to capture and leverage existing spatiotemporal dependencies.

**Results** We compare CoREL against standard SCP and the CoRNN variant. Moreover, we also compare performance against CoREL with access to the true graph used to generate the data. Results are shown in Tab. 2. When using an RNN as a point predictor (**GPVAR-RNN**) CoREL significantly outperforms both standard SCP and CoRNN. Furthermore, CoREL achieves results that closely match those obtained with direct access to the ground truth graph, which shows the effectiveness of the proposed architecture in capturing latent relational dependencies. Note that, given the injected Gaussian noise with $\sigma = 0.4$, the theoretical optimum PI width to obtain (asymptotically) 90% marginal coverage is 1.315. CoREL achieves essentially perfect coverage with PI width close to the theoretical optimum, while CoRNN requires a substantially higher PI width to obtain similar coverage. Finally, results obtained by using an STGNN as baseline (**GPVAR-STGNN**) show (as expected) that the standard SCP is sufficient when the point-predictor captures all the relevant dependencies. We provide a visualization of the learned graph in App. G.1.

Table 3: Adaptation results for **CER-E** for a reference CoREL model. ($\alpha = 0.1$).

| | RNN | | | Transformer | | | STGNN | | |
|---|---|---|---|---|---|---|---|---|---|
| | $\Delta$Cov | PI-Width | Winkler | $\Delta$Cov | PI-Width | Winkler | $\Delta$Cov | PI-Width | Winkler |
| W/o adapt. | -3.44 $\pm$ 0.40 | 1.84 $\pm$ 0.02 | 3.71 $\pm$ 0.03 | -3.08 $\pm$ 0.15 | 1.84 $\pm$ 0.02 | 3.69 $\pm$ 0.04 | -4.07 $\pm$ 0.47 | 1.80 $\pm$ 0.03 | 3.78 $\pm$ 0.03 |
| W/ adapt. | -2.70 $\pm$ 0.27 | 1.85 $\pm$ 0.01 | **3.49** $\pm$ **0.02** | -2.42 $\pm$ 0.20 | 1.85 $\pm$ 0.01 | **3.49** $\pm$ **0.04** | -3.06 $\pm$ 0.25 | 1.81 $\pm$ 0.02 | **3.58** $\pm$ **0.02** |

## 5.3. Adaptation

In this experiment, we evaluate how effectively the adaptation technique proposed in Sec. 3.4 provides accurate PIs over time. We focused on the CER-E dataset where the calibration set does not cover a full year, which likely introduces a shift at test time given the seasonality of energy consumption. We used CoREL with a fixed hyperparameter configuration across scenarios and trained on the calibration set. Embeddings are updated every $M$ time steps by running the training procedure on the latest observations and keeping all the parameters frozen except for the embeddings. More details on the hyperparameters are provided in App. B. In practice, this is done by splitting the test set into $K = 6$ folds and then iteratively fine-tuning the model on each fold. This procedure simulates a real-world scenario where new data become available over time and are used for fine-tuning. Results, reported in Tab. 3, show that this adaptation scheme improves performance by reducing the coverage gap and providing more accurate PIs.

## 6. Conclusion

In this paper, we introduced *Conformal Relational Prediction* (CoREL), a novel CP method for correlated time series. CoREL exploits graph-based neural operators to implement an uncertainty quantification architecture that can operate on top of any pre-trained point predictor. Furthermore, our approach does not require the relational structure to be known in advance. Results show that the proposed method compares favorably against the state-of-the-art in several relevant scenarios.

**Limitations and future work** We believe that CoREL constitutes an important step toward effective spatiotemporal CP methods. There are several directions for future work to explore, such as applying the framework to heterogeneous time series. Future work should focus on mitigating challenges arising from learning in non-stationary environments, e.g., by developing methods and coverage guarantees under specific assumptions about the non-stationarity of the data-generating process. Applications of CoREL to very large time series collections and combinations thereof with scalable graph-based architectures (Chiang et al., 2019; Cini et al., 2023c) are an additional direction. Finally, while CoREL accounts for dependencies w.r.t. past observations at correlated time series, it provides separate PIs for each time series. It would then be interesting to extend the framework with components that model the joint probability distribution of the time series.

## Impact Statement

This paper presents work whose goal is to advance the field of machine learning and time series forecasting. There are many potential societal consequences of our work, none of which we feel must be specifically highlighted here.

## Acknowledgments

This work was supported by the Swiss National Science Foundation grant no. 225351 (*Relational Deep Learning for Reliable Time Series Forecasting at Scale*) and no. 204061 (*HORD GNN: Higher-Order Relations and Dynamics in Graph Neural Networks*), the Hasler Stiftung project *Calibrated Uncertainty Estimation for Spatio-Temporal Data*, the UKRI CDT in AI for Healthcare (grant no. P/S023283/1), and the Norwegian Research Council grant no. 345017 (*RELAY: Relational Deep Learning for Energy Analytics*). FMB wishes to thank Nvidia Corporation for donating some of the GPUs used in this project. AC conducted part of this work while at the University of Oxford.

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

# Appendix

## A. Proof of Prop. 3.1

As stated in Prop. 3.1, let $P_{t+H}^c$ and $P_\psi^c$ indicate the probability distributions associated with the true data distribution at $t + H$ and with the learned quantile function $\mathcal{Q}_\psi$, respectively. It follows from the definition of total variation distance $TV(P, Q) := \sup_B |P(B) - Q(B)|$, that for any event $B$

$$|P_{t+H}^c(B) - P_\psi^c(B)| \leq TV(P_\psi^c, P_{t+H}^c) \quad \Longrightarrow \tag{20}$$

$$P_{t+H}^c(B) \geq P_\psi^c(B) - TV(P_\psi^c, P_{t+H}^c). \tag{21}$$

For $\hat{C}_{i,t}^\alpha$ in both Eq. 14 and Eq. 16 we have by construction that

$$P_\psi^c\left(\boldsymbol{x}_{t+H}^i \in \widehat{C}_{i,t}^\alpha\right) = 1 - \alpha. \tag{22}$$

Then, putting together Eq. 21 and Eq. 22 we obtain

$$P_{t+H}^c\left(\boldsymbol{x}_{t+H}^i \in \widehat{C}_{i,t}^\alpha(\widehat{\boldsymbol{X}}_{t+h})\right) \geq 1 - \alpha - TV\left(P_\psi^c, P_{t+H}^c\right).$$

Hence, the proof is complete.

## B. Hardware and software platforms

Benchmarks have been developed with Python (Van Rossum and Drake, 2009) and the following open-source libraries:

- Numpy (Harris et al., 2020);

- PyTorch (Paszke et al., 2019);

- PyTorch Lightning (Falcon and The PyTorch Lightning team, 2019);

- PyTorch Geometric (Fey and Lenssen, 2019);

- Torch Spatiotemporal (Cini and Marisca, 2022).

Experiments were conducted on a server equipped with AMD EPYC 7513 CPUs and NVIDIA RTX A5000 GPUs. The code for reproducing the computational experiments is available at `https://github.com/andreacini/corel`. For the HopCPT, SPCI, NexCP, and SeqCP baselines we use the open-source implementation made available by Auer et al. (2023)[1]. In our setup for the experiment in Sec. 5.1, considering the METR-LA dataset and the baselines that require fitting a model, training and testing require $\approx$ 3 days for SPCI, $\approx$ 11 hours for HopCPT, and $\approx$ 5 minutes for COREL.

## C. Datasets

We considered several datasets from different application domains, real-world scenarios and a simulated controlled environment. For the real world datasets we followed the pre-processing steps adopted by Cini et al. (2023b).

**Air Quality Monitoring**   The **AQI** (Zheng et al., 2015) dataset consists of 8,760 hourly measurements of pollutant PM2.5 from 437 monitoring stations in China. We use a window of $W = 24$ time steps and predict the 3-time-step-ahead observations. For the STGNN base model, we build an adjacency matrix using a thresholded Gaussian kernel computed from pairwise geographic distances (Shuman et al., 2013).

**Traffic Forecasting**   We considered the **METR-LA** (Li et al., 2018) traffic forecasting dataset, consisting of 34,272 timesteps of measurements from 207 loop detectors sampled at 5-minute intervals in the Los Angeles County Highways. We use a window of 12 time steps and predict the 12-time-step-ahead observations. For the STGNN base model, we followed previous works (Wu et al., 2019) and built an adjacency matrix using a thresholded Gaussian kernel applied to geographic distances.

---

[1] `https://github.com/ml-jku/HopCPT`

**Electric Load Forecasting**   We selected the **CER-E** dataset (Commission for Energy Regulation, 2016; Cini et al., 2022), comprising 25,728 timesteps of energy consumption readings aggregated at 30-minute intervals from 485 smart meters monitoring small and medium-sized enterprises. We use a window of $48$ time steps and predict the 5-time-step-ahead observations. For the STGNN base model, we built the adjacency matrix by extracting a 10-nearest neighbor graph from week-wise correntropy similarities between time series, following previous work (Cini et al., 2022).

**GPVAR**   For the GPVAR dataset, we generate synthetic data with 40,000 timesteps over an undirected network of 60 nodes connected in a community graph structure by following the system model in Eq. 19 (Zambon and Alippi, 2022). The parameters of the spatiotemporal process are set as

$$\Theta = \begin{bmatrix} 2.5 & -2.0 & -0.5 \\ 1.0 & 3.0 & 0.0 \end{bmatrix}, \quad a = b = 0.5, \quad \sigma = 0.4. \tag{23}$$

We used an input window of $5$ time steps to predict the next observation. For the STGNN base model, we used the same community graph structure as the adjacency matrix.

**Base models**   We trained three base models (point predictors) for each dataset: a RNN with GRU cells (1 layer with hidden size 32), a decoder-only Transformer (hidden size 32, feed-forward size 64, 2 attention heads, 3 layers, dropout 0.1), and a STGNN following the template in Sec. 2.4 (hidden size 32, node embedding size 16, 1 layer GRU, 2 message-passing layers). All models were trained by minimizing the MAE loss using the Adam optimizer for 200 epochs with batch size 32, using 40% of the data for training, 40% for calibration, and 20% for testing. We also use the first $25\%$ of the calibration data as a validation set for early stopping. Input features were scaled using standard scaling across time series.

## D. Additional details on CoReL implementation

**Architecture**   As discussed in Sec. 3 and Sec. 5, we implemented CoReL as TTS model with a single-layer GRU followed by 2 message passing layers analogous to those in (Satorras et al., 2022). As a readout, we used an MLP mapping the learned representations to predictions of 39 equally spaced quantiles. For the CoRNN baseline, we used the same architecture but removed the message passing layers and node embeddings $V$ and allowed the GRU to have more than one layer.

**Latent graph learning module**   The graph learning module was implemented as described in Sec. 3.2 by parametrizing $\Phi$ with a matrix of $N \times N$ learnable parameters. To allow for sampling less neighbors than the specified neighborhood size $K$, we followed previous works (Cini et al., 2023c) and modified the sampling procedure by introducing a set of dummy nodes then discarded from the sampled graph before message passing. We use dummy nodes in the GPVAR experiment only. To allow for sparse message-passing operations we use a straight-through gradient estimator (Bengio et al., 2013) and backpropagate gradients only through the sampled edges for each node, plus $10\%$ of the remaining ones. In the GPVAR experiment, we simply propagate gradients w.r.t. the entire adjacency matrix.

**Covariates**   Besides residuals, we use as additional covariates datetime encodings whenever available, plus the value of the target time series w.r.t. the time steps in the input window.

## E. Evaluation metrics

For a prediction interval $\widehat{C}_i^\alpha = [\hat{x}_i + \hat{q}_i^{\alpha/2}, \hat{x}_i + \hat{q}_i^{1-\alpha/2}]$ and true value $x_i$, evaluation is conducted for a desired confidence level $\alpha$ using three key metrics.

**Coverage gap**   $\Delta\text{Cov}_i$ measures the difference between the achieved coverage and the target coverage $1 - \alpha$, and is given by

$$\Delta\text{Cov}_i = 100\left(\mathbb{1}(x_i \in \widehat{C}_i^\alpha) - (1 - \alpha)\right), \tag{24}$$

where $\mathbb{1}$ denotes the indicator function.

**Prediction interval width**   Quantifies the width of the prediction intervals, and is given by

$$\text{PI-Width}_i = \hat{q}_i^{1-\alpha/2} - \hat{q}_i^{\alpha/2}. \tag{25}$$

**Winkler Score** Combines interval width with a penalty for predictions that fail to capture the true value, with misses penalized by a factor of $\frac{2}{\alpha}$. The Winkler score is given by

$$W_i = \begin{cases} (\hat{q}_i^{1-\alpha/2} - \hat{q}_i^{\alpha/2}) + \frac{2}{\alpha}(\hat{q}_i^{\alpha/2} - x_i) & \text{if } x_i < \hat{q}_i^{\alpha/2}, \\ (\hat{q}_i^{1-\alpha/2} - \hat{q}_i^{\alpha/2}) & \text{if } \hat{q}_i^{\alpha/2} \le x_i \le \hat{q}_i^{1-\alpha/2}, \\ (\hat{q}_i^{1-\alpha/2} - \hat{q}_i^{\alpha/2}) + \frac{2}{\alpha}(x_i - \hat{q}_i^{1-\alpha/2}) & \text{if } x_i > \hat{q}_i^{1-\alpha/2}. \end{cases} \tag{26}$$

All the metrics are then averaged over all nodes and time steps within the specified set.

## F. Hyperparameters and experimental setup

Hyperparameters were tuned separately for each combination of base predictor and dataset on a validation set.

**CoREL** For CoREL we tuned the number of neurons in the STGNN with a small grid search on $10\%$ of the calibration data. We used the same model selection procedure for CoRNN but also tuned the number of GRU layers. For the experiments on real-world data, the model was trained for a maximum of 100 epochs on the calibration set. Each epoch consisted of a maximum of 50 mini-batches of size 64. We used the Adam optimizer (Kingma and Ba, 2015) with an initial learning rate of 0.003 and reduced by $75\%$ every 20 epochs. We used a fixed number of $K = 20$ neighbors for the graph learning module. For the GPVAR experiment in Sec. 5.2, we fixed the number of neurons to 16 for each layer and used an embedding size of 8. For the adaptation experiment in Sec. 5.3, we use a reference configuration with 64 neurons in each encoder/decoder layers and an embedding size of 32. We train the entire model on the full calibration set and use adaptation at test time. In particular, at test time, we fine-tune the node embeddings every $M$ time steps, where $M$ corresponds to $\frac{1}{6}$ of the test set length. For fine-tuning, we fit the embeddings by running 25 epochs of maximum 10 batches each by using samples from the last $M$ steps with a fixed learning rate of 0.001.

**HopCPT** For HopCPT we followed the model selection procudere described in (Auer et al., 2023). The model was trained for 3000 epochs using a batch size of 4 time series. We adopted the paper's AdamW optimizer configuration with standard parameters ($\beta_1 = 0.9$, $\beta_2 = 0.999$, $\epsilon = 0.01$) and tuned the model by running the same hyperparameter configurations searched by (Auer et al., 2023). All the remaining hyperparameters were set accordingly to the original paper.

**SeqCP** SeqCP, similarly to (Xu and Xie, 2023a), employs a sliding window approach to conformal prediction, assigning equal weights to observations within the most recent $K$ time steps and zero weights to older observations. The window size $K$ was treated as a hyperparameter and tuned over the values $\{200, 150, 125, 100, 75, 50, 25, 10\}$.

**NexCP** NexCP implements conformal prediction using exponentially decaying weights controlled by a parameter $\rho$. Rather than using a fixed window of historical observations, it assigns weights that decay exponentially with time, giving more recent observations higher importance. The decay parameter $\rho$ was tuned over the values $\{0.999, 0.995, 0.993, 0.99, 0.98, 0.95, 0.9\}$.

**SPCI** As already mentioned, we used the SPCI (Xu and Xie, 2023b) implementation provided by Auer et al. (2023) and followed an analogous protocol for training the mode. In particular, SPCI was run using a fixed window length of 100 time steps for all experiments, corresponding to the longest setting in the original paper. The computational demands of SPCI were substantial, as separate quantile random forests had to be trained for each combination of node and target coverage level, making extensive hyperparameter tuning impractical. To further manage the computational costs, we trained each SPCI model only once on the calibration data, rather than implementing the time-adaptive approach where models are re-trained as new observations become available.

## G. Additional results

### G.1. Qualitative analysis of the learned graph on GPVAR

Fig. 2 provides a comparison of the graph learned on GPVAR by CoREL in Sec. 5.2. In particular, the rightmost figure is obtained by taking the top-$K$ scores associated to each row[2]. The figure shows that the learned graph includes true edges

---

[2]The figure does not show edge scores associated to dummy nodes; see App. D

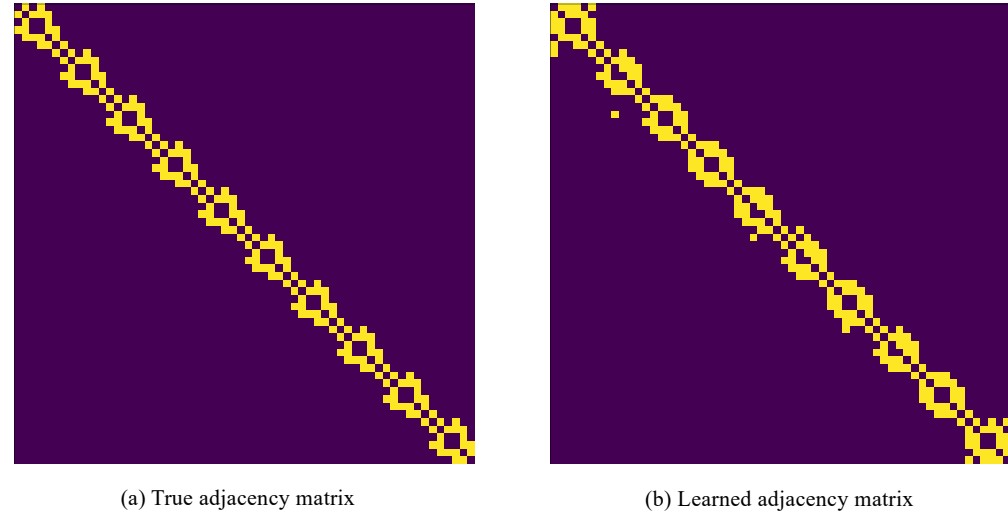

(a) True adjacency matrix  (b) Learned adjacency matrix

Figure 2: Comparison between **(a)** the true graph and **(b)** the graph structure learned by CoReL in GPVAR.

plus additional links. Note that perfectly recovering the ground truth graph is not required to achieve optimal performance here.

### G.2. Optimization of the PI width

Here, we show how the width of the PI can be reduced by searching for an appropriate quantile offset $\beta$. Specifically, we perform the optimization described in Eq. 16, which identifies pairs of quantiles yielding the same coverage $1 - \alpha$ while achieving the smallest interval width. We summarize our results in Tab. 4. As we can see, the procedure consistently finds intervals with a smaller width at the price of slightly reducing the coverage, whether using CoReL or the CoRNN variant.

Table 4: Changes in $\Delta$ Cov and PI-Width when optimizing $\beta$ ($\alpha = 0.1$).

| | | | RNN | | Transformer | | STGNN | |
|---|---|---|---|---|---|---|---|---|
| | | | $\Delta$Cov | PI-Width | $\Delta$Cov | PI-Width | $\Delta$Cov | PI-Width |
| METR-LA | CoRNN | | -0.70 $\pm$ 0.20 | 19.69 $\pm$ 0.27 | -0.52 $\pm$ 0.56 | 19.07 $\pm$ 0.42 | -0.67 $\pm$ 0.47 | 16.02 $\pm$ 0.13 |
| | | $\beta$ optim. | -0.80 $\pm$ 0.28 | 17.71 $\pm$ 0.28 | -0.96 $\pm$ 0.35 | 17.16 $\pm$ 0.46 | -0.88 $\pm$ 0.44 | 14.98 $\pm$ 0.17 |
| | CoReL | | -1.25 $\pm$ 0.80 | 14.24 $\pm$ 0.53 | -0.97 $\pm$ 0.51 | 14.43 $\pm$ 0.38 | -1.31 $\pm$ 0.48 | 14.36 $\pm$ 0.20 |
| | | $\beta$ optim. | -1.49 $\pm$ 0.56 | 13.23 $\pm$ 0.46 | -1.26 $\pm$ 0.42 | 13.47 $\pm$ 0.35 | -1.44 $\pm$ 0.45 | 13.46 $\pm$ 0.20 |
| CER-E | CoRNN | | -2.09 $\pm$ 0.56 | 1.93 $\pm$ 0.01 | -1.70 $\pm$ 0.37 | 1.99 $\pm$ 0.02 | -2.45 $\pm$ 0.41 | 1.85 $\pm$ 0.03 |
| | | $\beta$ optim. | -2.63 $\pm$ 0.65 | 1.82 $\pm$ 0.01 | -2.28 $\pm$ 0.31 | 1.88 $\pm$ 0.02 | -3.10 $\pm$ 0.52 | 1.78 $\pm$ 0.03 |
| | CoReL | | -3.86 $\pm$ 0.53 | 1.83 $\pm$ 0.02 | -3.81 $\pm$ 0.30 | 1.83 $\pm$ 0.02 | -4.87 $\pm$ 0.51 | 1.79 $\pm$ 0.04 |
| | | $\beta$ optim. | -4.16 $\pm$ 0.59 | 1.75 $\pm$ 0.02 | -4.10 $\pm$ 0.21 | 1.75 $\pm$ 0.02 | -5.13 $\pm$ 0.51 | 1.72 $\pm$ 0.03 |
| AQI | CoRNN | | -1.45 $\pm$ 0.61 | 73.88 $\pm$ 1.17 | -1.08 $\pm$ 0.22 | 74.02 $\pm$ 0.74 | -1.50 $\pm$ 0.63 | 70.42 $\pm$ 0.94 |
| | | $\beta$ optim. | -1.90 $\pm$ 0.54 | 70.17 $\pm$ 1.22 | -1.10 $\pm$ 0.25 | 70.95 $\pm$ 0.68 | -1.38 $\pm$ 0.62 | 68.24 $\pm$ 0.85 |
| | CoReL | | -2.40 $\pm$ 1.97 | 70.44 $\pm$ 2.69 | -3.54 $\pm$ 1.80 | 68.31 $\pm$ 4.49 | -2.61 $\pm$ 1.92 | 67.76 $\pm$ 3.12 |
| | | $\beta$ optim. | -2.83 $\pm$ 1.86 | 67.35 $\pm$ 2.31 | -4.19 $\pm$ 1.96 | 65.20 $\pm$ 4.20 | -3.22 $\pm$ 2.45 | 65.32 $\pm$ 3.43 |

