# OpenReview forum: "Relational Conformal Prediction for Correlated Time Series"
_ICML.cc/2025/Conference — ICML 2025 poster_

### Official Review · Reviewer_uUN5 · 2025-03-02

**Overall Recommendation:** 3

**Summary:**

This paper introduces a method for improving uncertainty quantification in time series forecasting by leveraging correlations between sequences. The authors propose Conformal Relational Prediction (COREL) that integrates conformal prediction and quantile regression with graph-based deep learning. The method captures relationships between time-series without requiring prior knowledge of the graph structure. It also adapts to non-exchangeable data and time series variations.

**Claims And Evidence:**

I appreciate the authors' effort in comparing their method against multiple baseline approaches across different datasets. However, the results in Table 1 are challenging to interpret. Notably, CoREL consistently fails to achieve the targeted marginal coverage across datasets. As a result, it is unclear whether the smaller prediction interval widths (or lower Winkler scores) reflect the superiority of the proposed method or simply stem from its anti-conservativeness.

**Essential References Not Discussed:**

Relevant related works are cited as far as I am aware.

**Experimental Designs Or Analyses:**

The method proposed by the authors learn the graph topology used in the GNN to estimate quantiles from the residuals on the calibration set. Nevertheless, the authors do not discuss the learned graph topology (e.g. from in interpretability viewpoint). I would have been also interesting to conduct an ablation study to see if learning the graph really helps compared to considering a

**Methods And Evaluation Criteria:**

The metrics and dataset considered to evaluate the method make sense for the application at hand. Nevertheless, the paper focus on the prediction interval widths, which is inherently due to the fact the method gives confidence regions (in R^N where N is the number of time series) which are cubes. In my opinion, this is one limitation of the method. Please refer to the section "Questions to for authors" for more details).

**Other Comments Or Suggestions:**

Here are some typos:

- Eq (9): h_t^L should be H_t^L.

- In Eq.(9), should it be \hat y^{i}_t  and $h^{i,L}_t$ ?

**Other Strengths And Weaknesses:**

-

**Questions For Authors:**

I would be grateful if the authors could clarify the following aspects:

- Why only the residuals are provided as input in the RelQPs and not providng also the history of the time-series themselves ?

- Could the authors comment on the motivation to feed to the decoder the time series independently rather than having still a single decoder but that would take as input the matrix Z_t to predict quantiles in Eq.(12)?

- The final conformal sets are cubes (cf. Eq. (14)). This form of conformal sets appears suboptimal, as it does not leverage the correlation structure learned by the RelQPs between the time series to produce a confidence region that adapts to the dependence structure of the data rather than remaining cubic.

- In the paper, the authors explain that "the relational structure is sparse" (between the N time series). The way this sparsity prior is used as an inductive bias in the model architecture and/or the optimization scheme is not very clear to me. It seems that this question is touched in Appendix D but there were not enough details for me to understand completely. Could the authors provide more information on this sparsity aspect ?

- The way the adaptive version of the method works is not clear to me. In Section 5.3, it is stated that "we split the test set in K = 6 folds and then iteratively evaluated and fine-tuned the model on each fold." How to use the method is practice (since in real aplication, the test set is in the future and we don't have measurements) ? In my understanding, people would need to split the calibration set into two parts: one used to train RelQPs and the used to fine-tune the learnable node embeddings V.

**Relation To Broader Scientific Literature:**

The paper is at the crossroad of several important topic in the field of uncertainty quantification. The propose method aims to quantify uncertainty when dealing with spatio-temporal data in a multivariate settings.
Most existing methods in the litterature on conformal prediction typically focus on one of these challenges (i.e. methods designed to deal with multivariate output data, or spatial data or non exchangeable data).

Nevertheless, the way the paper is written gives the impression that the introduced method is a conformal prediction approach. However, the method lacks a theoretical guarantee of non-asymptotic valid coverage. Therefore, claiming that it falls within the conformal framework is not justified, in my opinion.

**Theoretical Claims:**

There is no contribution on the theoretical side in the paper. Proposition 3.1 follows from the definition of the total variation distance.

---

> ### Author Rebuttal · Authors · 2025-03-31
>
> Thanks for the review, please find point-by-point answers below.
>
> >Unclear if smaller PIs (or lower Winkler) simply stem from anti-conservativeness.
>
> There might be a misunderstanding. The Winkler score (see Eq. 26 in the Appendix) encompasses both coverage and efficiency, i.e., a smaller PI width does not imply a lower Winkler score. Furthermore, CP methods cannot guarantee exact coverage in these scenarios due to non-exchangeability (see comment on guarantees below).
>
> > Discussion of the learned graph topology.
>
> Experiments on GPVAR and the inclusion of the CoRNN datasets go exactly in this direction. CoRel consistently outperforms the CoRNN baseline in Tab1 showing the effectiveness of the proposed designs. Results on GPVAR show that CoREL matches the performance achievable by accessing the ground truth graph and achieves UQ performance comparable to the theoretical optimum. We will also include visualizations of the learned graph (e.g., https://imgur.com/a/uJaYNnX) - see our answer to Rev oUbs for more discussion on these results on the GPVAR dataset.
>
> > No theoretical guarantees for non-asymptotic coverage: claiming that it falls within CP is not justified.
>
> Without any strong assumption, CP methods for TS forecasting cannot offer non-asymptotic coverage guarantees as TS data are, in general, not exchangeable (see [1]). As discussed in our paper, most CP methods for TS rely on learning either a model [2,4] or a reweighting scheme [1, 3]. The same is true for adaptive methods such [5].  CoRel belongs to the family of methods that rely on a model trained on non-conformity scores; it is distribution-free and can be applied on top of any base point predictor.
>
> > Typo eq. 9
>
> Thanks, the correct version is:
>
> $$\hat y^i_{t} = Readout \left(h_t^{i,L}\right)$$
>
> > Q1 Why only residuals as input and not also the history of the TS?
>
> There might be a misunderstanding, as stated in lines 195–198 (right), we also use the actual time series as part of the input.
>
> > Q2 Motivation to feed to the decoder TS independently rather than matrix Z_t in Eq.(12).
>
> Because: (1) it would result in a fully connected layer with a large number of learnable parameters likely to incur overfitting; (2) it would constrain the model to operate on the full set of TS at once while the proposed architecture can operate on any subset separately (more scalable); (3) it would defeat the purpose of using a message-passing architecture rather than a simple fully connected MLP; (4) while the UQ procedure accounts for past observations at correlated TS, prediction intervals for each TS are independent of the others (see below).  Note that analogous parameter-sharing architectures, with message passing and sequence modeling operators, are at the core of graph-based processing for TS [6].
>
> > Q3 Conformal sets are cubes (cf. Eq. (14)). This appears suboptimal, as it does not leverage the learned structure.
>
> There might be a misunderstanding, which might be due to the admittedly dense notation (we will make sure to improve it). Our framework leverages the learned correlation structure to account for dependencies w.r.t. **past** observations at correlated TS rather than to model the joint distribution of future observations. In particular, we model conditional probabilities as in Eq. 1 (i.e., we model $p(x_t^i|X_{<t}, U_{<t}) \forall i$ rather than $p(X_t|X_{<t}, U_{<t}))$. While modeling joint probabilities within our framework can be interesting, it is a different problem out of scope here. We will clarify this aspect and mention it as an important direction. Thanks for raising this point.
>
> > Q4 The way this sparsity prior is used is not clear to me.
>
> As explained in Sec. 3.2 we parametrize the graph learning module to learn at most K<<N edges for each node. This improves scalability and can also improve sample efficiency [7].
>
> > Q5 The way the adaptive version of the method works is not clear to me.
>
> We train the entire model on the full calibration set and use adaptation at test time. Basically, at test time, we simulate a real-world scenario where new data become available over time and are used for fine-tuning. Instead of retraining the entire model (as in [2]), we just fine-tune the node embeddings. Instead of finetuning at each time step (high variance and high computational cost), we fine-tune the model every M time step.
>
> [1] Barber et al. “Conformal prediction beyond exchangeability” Ann. Statist. 2023\
> [2] Xu et al. “Sequential Predictive Conformal Inference for Time Series” ICML 2023\
> [3] Auer et al. “Conformal Prediction for Time Series with Modern Hopfield Networks” NeurIPS 2023\
> [4] Xu et al. ”Conformal prediction interval for dynamic time-series” ICML 2021\
> [5] Gibbs et al. “Adaptive conformal inference under distribution shift” NeurIPS 2021\
> [6] Cini et al. “Graph deep learning for time series forecasting” arXiv 2023\
> [7] Cini et al. “Sparse graph learning from spatiotemporal time series” JMLR 2023

---

### Official Review · Reviewer_oUbs · 2025-03-12

**Overall Recommendation:** 3

**Summary:**

The work introduces Conformal Relational Prediction (CoREL), a novel distribution-free uncertainty quantification method for time series forecasting that leverages graph deep learning. CoREL integrates spatiotemporal graph neural networks with conformal prediction (CP) to capture relational structures among correlated time series and estimate prediction intervals more effectively.

## update after rebuttal
Thank you to the authors for the rebuttal. Most of my concerns have been adequately addressed and I feel it a good combination about graph representations with time series, so I have decided to raise my score.

**Claims And Evidence:**

The work provides strong empirical support for several key claims. CoREL’s ability to capture relational structure for uncertainty quantification is demonstrated through both real-world datasets and a controlled synthetic experiment, confirming that the learned relationships enhance prediction intervals. Additionally, CoRELs adaptive learning mechanism is tested on non-stationary data, showing that it can adjust dynamically to evolving time series patterns.
However, as the paper claims that CoREL scales well and is computationally efficient. The method requires graph inference at every update step, which may introduce non-trivial overhead. Comparisons against standard CP methods in terms of runtime are needed to fully validate efficiency claims.

**Essential References Not Discussed:**

No.

**Experimental Designs Or Analyses:**

The experimental design of CoREL demonstrates several strengths, particularly in its comprehensive evaluation across multiple real-world time series datasets. The inclusion of diverse baselines ensures that CoREL's performance is rigorously benchmarked against existing conformal prediction methods. The study also examines the adaptability of CoREL in non-stationary settings by performing cross-validation and fine-tuning, demonstrating its ability to refine uncertainty estimates dynamically.

**Methods And Evaluation Criteria:**

The methods of leveraging GNN to capture relational structures in time series make sense overall. The evaluation criteria in experiments such as the coverage rate and prediction interval width aligns with conformal prediction and prior works.

**Other Comments Or Suggestions:**

The paper is well-structured, clear, and easy to follow. While there are some weaknesses, they do not significantly impact the overall quality.

**Other Strengths And Weaknesses:**

Strengths:
1. CoREL effectively integrates conformal prediction with graph learning, capturing dependencies in correlated time series. It avoids strong distributional assumptions and dynamically adapts to non-stationary data.

2. The model infers inter-series relationships, eliminating the need for predefined graphs.

3. Comprehensive Evaluation – Tested across diverse datasets with ablation studies, reinforcing its empirical validity.

Weaknesses:
1. Graph inference and message passing may introduce scalability issues.

2. While the model dynamically learns dependencies, there is no formal guarantee that the learned graph structure is optimal for uncertainty quantification, though it proves effective in experiments. Visualizations or comparisons could strengthen this aspect.

3. While CoREL is compared to standard CP methods, it does not benchmark against other graph-based uncertainty quantification approaches, such as Bayesian graph neural networks or CP for GNNs [1]. It would be interesting to compare and discuss the results here.

[1] Zhao, T., Kang, J., & Cheng, L. (2024, August). Conformalized link prediction on graph neural networks. In Proceedings of the 30th ACM SIGKDD Conference on Knowledge Discovery and Data Mining (pp. 4490-4499).

**Questions For Authors:**

1. Please refer to weaknesses.

2. I am not deeply familiar with the specifics of the datasets, but I wonder if they exhibit strong correlations. Are there any limitations to CoREL? Under what conditions might its performance degrade? A discussion on the method’s limitations would be valuable.

**Relation To Broader Scientific Literature:**

CoREL contributes to the broader scientific landscape by bridging conformal prediction, graph-based deep learning, and uncertainty quantification in structured data. Traditional conformal methods assume independence, limiting their effectiveness in relational settings. By dynamically learning inter-series dependencies, CoREL aligns with advances in spatiotemporal modeling, probabilistic inference, and adaptive learning in graph-based frameworks. Its approach connects with broader efforts in representation learning, self-supervised graph modeling, and uncertainty-aware neural architectures, highlighting the growing need for flexible, data-driven methods that generalize across domains.

**Theoretical Claims:**

The theoretical guarantee for Proposition 3.1 is correct.

---

> ### Author Rebuttal · Authors · 2025-03-31
>
> Thank you for the review. Please find our comments below.
>
> > Comparisons against standard CP methods in terms of runtime are needed to fully validate efficiency claims.
>
> The main two baselines to compare to here would be SCPI and HopCPT. Scalability issues for SCPI comes from training a different model for each time series which results in high computational costs when dealing with multiple TS. Regarding HopCPT, the main computational bottleneck is due to attention scores being computed w.r.t. the considered sequence at each time step which results in a quadratic complexity w.r.t. the sequence length. Conversely, CoRel shares most of the parameters among the time series collection and its training can be efficiently parallelized on a GPU. Furthermore, it relies only on a short window of the most recent observations + node embeddings at each time step. If we consider LA (which is the smallest dataset in terms of numbers of time series) training and testing require **~3 days for SCPI , ~11 hours for HopCPT and ~5 minutes for CoRel**. We will put more emphasis on this aspect in the paper.
>
> > W1 Graph inference and message passing may introduce scalability issues.
>
> Scalability is inherently an issue when dealing with multiple TS simultaneously, so it can be a challenge, but not a limitation of our method. Additionally, graph inference and message-passing layers can have different computational complexity, depending on how the different components are implemented. Research on scalable graph structure learning methods (e.g., [1]) and STGNN architectures is very active (e.g., see [2,3] for discussion on existing architectures); these approaches could be integrated in the framework. Furthermore, one can rely on graph subsampling techniques (e.g., [4]) to enable scalability. We will mention these points in the paper, thank you.
>
> > W2 No formal guarantee that the learned graph structure is optimal[...] Visualizations or comparisons could help.
>
> First of all, there is not a clear optimality criterion here as the graph is directly learned end-to-end with the quantile regressor and as such it will strongly depend on the actual implementation of the quantile network (e.g., no. of layers, no. of message-passing steps, and so on). Learning a graph here serves the purpose of obtaining good performance on the downstream task. That being said, the experiment on the synthetic GPVAR dataset (where the ground-truth graph is known a priori) shows that CoRel can match the performance of the model that has access to the true graph and in this scenario, the learned graph closely matches the ground truth. A visualization of the learned graph against the ground truth is available here:  https://imgur.com/a/uJaYNnX. As shown by the picture the learned graph includes all the actual edges plus some additional links.  As detailed in the paper, in GPVAR, data were created by injecting Gaussian noise into the diffusion process with a std of 0.4 which implies that having direct access to the data-generating process would allow one to obtain (asymptotically) a 90% marginal coverage with a PI width of **1.315**. CoRel achieves essentially perfect coverage with PI width of **1.329 $\pm$ .002 (close to the theoretical optimum)** while CoRNN requires a PI width of **1.63 $\pm$ 0.01** to obtain similar coverage. We will include these comments and visualizations of the learned adjacency matrix in the paper. Thank you for raising this point.
>
> > W3 No benchmark against other graph-based uncertainty quantification approaches such as [1]. It would be interesting to compare and discuss the results here.
>
> We discuss several graph-based approaches for uncertainty quantification in the related works, but, as noted in the paper, none of these are post hoc methods and, as such, cannot be directly compared to our approach. Existing CP approaches for graphs are also not applicable to TS processing, e.g., the referenced paper targets link prediction. Nonetheless, we agree that the discussion on this aspect can be extended further, and we will include [1] as a relevant reference.
>
> > Q2 Not deeply familiar with the specifics of the datasets [...] Are there any limitations to CoRel? Under what conditions might its performance degrade? A discussion on limitations would be valuable.
>
> The considered datasets are well-known in correlated TS forecasting and have been extensively used in the related literature. CoRel relies on the assumption of the existence of a Granger causality among TS (as mentioned in the paper), if that is not the case we can expect its performance to be analogous to that of the CoRNN baseline. Indeed, as shown in the experiments, the gap in performance between CoRel and CoRNN is smaller when using an STGNN as a base model, as the STGNN already partially captures relational dependencies. In the updated paper, we will comment on these aspects and provide additional details on future work (see rebuttals to other reviewers).

---

### Official Review · Reviewer_oqTA · 2025-03-14

**Overall Recommendation:** 4

**Summary:**

The paper presents Conformal Relational Prediction (COREL), which integrating graph deep learning (GDL) operators into the CP framework, allowing relational structures to improve uncertainty estimation for spatio-temporal time series. The method utilizes a STGNN to provide structural embeddings and applies quantile regression to produce the PIs. The algorithm can be modified to become adaptive to data shifts (by modifying local parameters). Strong experimental results supports authors claims.

**Claims And Evidence:**

Yes, the claims are well-supported.

**Essential References Not Discussed:**

should also cite the original conformal quantile regression paper:

@article{romano2019conformalized,
  title={Conformalized quantile regression},
  author={Romano, Yaniv and Patterson, Evan and Candes, Emmanuel},
  journal={Advances in neural information processing systems},
  volume={32},
  year={2019}
}

**Experimental Designs Or Analyses:**

Comprehensive baseline selection, good metrics, nice diverse datasets. Experiment section is well done.

**Methods And Evaluation Criteria:**

Great, no notes.

**Other Comments Or Suggestions:**

typos:

- abstract line 38: "archives state-of-the-art uncertainty quantification" → "achieves state-of-the-art uncertainty quantification".
- Method by Xu & Xie 2023b is SPCI (sequential predictive conformal inference) instead of SCPI...

**Other Strengths And Weaknesses:**

Strong paper for uncertainty quantification. Good method, well-motivated and well-presented.

**Questions For Authors:**

Prop 3.1 is the main theoretical support for the method. However, it blankets everything that could undermine the conformal guarantee under the total variation. This is not great because in reality it's hard to estimate or control TV (since it is the sup over the entire space). So there is really no guarantees other than "let's hope our quantile regression works well".

The authors wrote, following prop 3.1
> By making assumptions on the expressivity of the quantile regressor in Eq. 13 and on the stationarity process (e.g., by assuming a strongly mixing process), we can expect the total variation between the learned and true distribution to shrink asymptotically as the size of the calibration set increases

but provided no proof to support this claim. (I know this is a common argument, but even supporting citations will help, a proof with the specific setup will be better. ) Most previous conformal work that incorporates quantile regression, from CQR to  SPCI and [this paper](https://openreview.net/forum?id=oP7arLOWix) (also) by Lee, Xu and Xie, contains an actual validity proof.

**Relation To Broader Scientific Literature:**

GNN structure-informed conformal quantile regression has not been explored before in literature. Given the importance of GNN in modeling Spatiotemporal data, this is very useful, as demonstrated by the authors via experiments.

**Theoretical Claims:**

Correct but not the most illuminating. See question section.

---

> ### Author Rebuttal · Authors · 2025-03-31
>
> Thank you for the review! Please find our answers to your questions below.
>
> > Missing reference.
>
> We agree it is a relevant reference. We will include it in the updated version of the paper. Thank you.
>
> > Typos.
>
> Thank you for spotting those!
>
> > Prop 3.1 [...] blankets everything that could undermine the conformal guarantee under the total variation. [...] The authors claim that [we can expect the total variation between the learned and true distribution to shrink asymptotically] but provided no proof to support this claim. [...] this is a common argument, but even supporting citations will help, a proof with the specific setup will be better.
>
> Thank you for the comment. As noted, a proof would require assumptions on the data generation process and on the specific implementation of the quantile regressor and message-passing operators. We feel this would not add a lot to the paper since, as you noted, a similar analysis has been done in recent works by Lee, Xu, and Xie. We do, however, agree on the necessity to further discuss this aspect and the related references; we will do this in the updated version. Thanks again for the comment.

---

### Official Review · Reviewer_aHCK · 2025-03-23

**Overall Recommendation:** 3

**Summary:**

This work is on uncertainty quantification in time series forecasting. The authors proposed a conformal prediction method based on graph neural networks. Their approach is based on quantile regression. A spatiotemporal graph neural network was trained on the residuals of the calibration dataset to predict the quantiles of the error distribution. The predicted residuals are used to learn the quantile function of the error distribution at each time step. The performance metrics used were: the difference between the specified confidence level and the observed coverage on the test set; the width of the PI and the Winkler score. The proposed approach was tested on three benchmark datasets and the proposed model outperforms competitors in terms of Winkler score for most of the test cases. Its performance is comparable to other models based on the other two metrics.

## update after rebuttal
The explanations provided by the authors and the changes they intend to make are satisfactory. However, I would stay with the rating recommended earlier.

**Claims And Evidence:**

The results presented show that the proposed model is better than other models based on Winkler score but other models perform as well or better based on the other two metrics. Hence, once cannot completley claim that the proposed model is superior to previous models.

**Essential References Not Discussed:**

The main contribution of this work in the development of STGNN for conformal prediction. The authors cited relevant literature and provided appropriate references. Fairly decent review of the previous work was provided in Section 4. The authors identified the shortcomings of the previous works in terms of STGNNS and CP and cited relevant literature.

**Experimental Designs Or Analyses:**

I did not see any experimental designs or analyses of the same or results of such experimental designs. The authos have essentially tested their proposed method on three standard benchmarks and presented the results.

**Methods And Evaluation Criteria:**

The proposed method and the evaluation criteria are appropriate. But, since the authors claim that this work is mainly addresses the spatio-temporal relationships in multivariate time series data, it would have been better if they clearly demonstrated the value of this method with detailed explanation and results at least for one of the test cases, showing how taking spatio-temporal relationships has provided additional insights into the behavior of the system.

**Other Comments Or Suggestions:**

It is important to quantitatively demonstrate the value of conformal prediction with quantile regression over forecasting target variables. The comparison has to be on the approach rather than the STGNN model.
Secondly, it would be better if the authors take a real life example, and demonstrate the value of the work rather than the standard performance metrics showing improvement with a few percent points or improvement in third or fourth decimal. Such works do not really make any difference to practising engineers.
Similarly, since this work is on STGNN, it's important to demonstrate the true value with a multivariate time series forecasting test case and provide insights on how incorporating dynamic spatio-temporal realtionships make a difference in uncertainty estimation.

**Other Strengths And Weaknesses:**

Both STGNNs and conformal prediction methods exist. But where the current work comes out good is in introducing quantile regression for predicting the quantiles of the error distribution of an existing model, rather than on forecasting the target variable. The proposed approach appears to be more adaptive. The local components of the model are updated over time while retaining the global relationships. However, the authors focused only on univariate analysis. Ideally, one should demonstrate the approach with multivariate time series. Secondly, while this approach is different from previous works, it's important to demonstate how predicting the quantiles of the error distribution is better than forecasting the target variables. The advantage has to be shown quantitatively.
The authors demonstrated the approach on three benchmarks and provided the standard test results. One of the main drawbacks of this work as well as similar such works is, one does not know actual utility of this work to an application engineer. How does this work make a difference in real life? Would a one or two pecent improvement in Winkler score make a great difference to uncertainty estimation and can it alter final decisionmaking? Are the improvements shown in performance metrics with one or two percent improvements, and in some cases, just in the third or fourth decimal point, make any difference to the end user?

**Questions For Authors:**

Please see the earlier comments.

**Relation To Broader Scientific Literature:**

This work could be considered an incremental contribution to uncertainty estimation in multivariate time series forecasting. It is certainly of value but needs further testing and evaluation in detail for specific use cases to see if it has real practical utility.

**Theoretical Claims:**

The authors made a theoretical proposition and also provided a short proof for the same (Section 3.3 and Appendix A)

---

> ### Author Rebuttal · Authors · 2025-03-31
>
> Thank you for your review! Please find our point-by-point answers below.
>
> > Results show that the proposed model is better on Winkler score but not on the other two metrics. Hence, cannot claim that the proposed model is superior.
>
> Coverage and PI Width on their own do not say much about UQ  performance, as one can trivially get high coverage with an arbitrarily large prediction interval or get small PI Width with a narrow, but not valid, one. Doing well on both aspects at the same time is the challenge: the Winkler score (see Appendix) considers coverage and PI width at the same time. We will add this comment to the paper.
>
> > It would have been important to show how modeling spatio-temporal relationships has provided additional insights.
>
> The inclusion of the CoRNN baseline which does not include any message passing or graph learning component, provides the insights asked by the reviewer by showing the impact of the proposed designs.  Furthermore, experiments on the synthetic GPVAR dataset (where spatial dependencies determine the observed dynamics) show that CoRel can match the performance of a model that has access to the ground-truth graph and obtain UQ performance close to the theoretical optimum (see rebuttal to Rev oUbs). Additionally, there is a wide literature on how modeling and learning relational dependencies is beneficial in TS (e.g., see [1,2]). Our framework allows one to benefit from such representations in post hoc UQ.
>
> > I did not see any experimental designs or analyses of the same or results of such experimental designs.
>
> There might be a misunderstanding. We designed a set of benchmarks for assessing post hoc UQ quantification on correlated TS. The selected datasets have already been used in the context of TS forecasting, but their use in the context of UQ is new. This also required setting up different base models to test UQ methods in different scenarios for each dataset. Furthermore, we also included a synthetic dataset to further validate the proposed designs. We believe that experiments do show the effectiveness of CoRel.
>
> > The authors focused on univariate analysis. Ideally, one should demonstrate the approach with multivariate time series.
>
> Our approach can be extended to collections of multivariate TS by pairing the proposed framework with a quantile regressor able to handle multivariate data at each node, e.g., [3], or by simply using a separate quantile regressor for each output channel. Note, however, that this is orthogonal to the proposed approach. We will include a comment on this in the paper.
>
> > While the approach is different from previous works, it's important to quantitatively demonstrate how predicting the quantiles of the error distribution is better than forecasting the target.
>
> There might be a misunderstanding. As with any CP method, CoRel quantifies the uncertainty of the predictions of a base pre-trained model. This is a fundamentally different problem from learning a probabilistic predictor directly. It is not possible to set up a direct comparison as the results would heavily depend on the base predictor. This difference is pointed out in Sec. 1 and 4, we will make sure to further emphasize this aspect.
>
>
> > One does not know actual utility of this work to an application engineer. How does this work make a difference in real life?
>
> We agree on the importance of engineering applications. The usefulness of more accurate UQ (and CP methods in particular) in real-world applications is widely recognized [4]. However assessing the practical impact of a particular method in a specific application and downstream task is orthogonal to our contribution, which is on fundamental research in ML and statistics.
>
> > Would a 1 or 2 %improvement in Winkler score make a great difference? Any difference for the user?
>
> There may be a misunderstanding. Differences in Winkler score are quite significant in all the considered scenarios (even more than 10% in some cases) and a 10% difference in performance is highly significant in any engineering application. As discussed in previous answers, coverage and PI width should be considered together. For an example of how CoRel's prediction intervals look compared to the CoRNN baseline in a given scenario (LA dataset with an RNN base model), see https://imgur.com/a/UYIj0nR.  Additionally, we believe that the consistent performance improvements and methodological novelty make our paper a relevant contribution on its own, besides specific applications. The analysis of specific applications in science and engineering is out of scope and would require separate studies.
>
> [1] Jin et al. “A survey on graph neural networks for time series” TPAMI 2024\
> [2] Cini et al. “Graph deep learning for time series forecasting” arXiv 2023\
> [3] Feldman et al. “Calibrated Multiple-Output Quantile Regression with Representation Learning” JMLR 2023\
> [4] Smith, “Uncertainty Quantification: Theory, Implementation, and Applications” SIAM 2013

---

> > ### Comment · Reviewer_aHCK · 2025-04-05
> >
> > While I agree with the explanations provided by the authors, the inherent weaknesses of the work still remain, for ex., demonstration with univariate data. The theoretical contribution is only marginal and not giving enough importance to demonstration with real-life examples. Hence, the overall recommendation will remain the same.

---

> > > ### Author Response · Authors · 2025-04-05
> > >
> > > Thank you for the comment.
> > >
> > > We genuinely respect your opinion, but we think the paper already makes relevant contributions and addresses many aspects, e.g., 1) conformal prediction on collections of correlated time series, 2) graph-based quantile regression from residuals, 3) probabilistic latent graph learning, 4) hybrid global-local time series processing, 5) adaptation and fine-tuning of a global-local UQ model and more. Given the necessity of keeping the scope of a conference paper contained, we feel that addressing real-world applications more than what we already do (we use datasets coming from 3 different very practical applications--traffic, air quality, and energy analytics) would dilute the content of the paper too much. Similar comments can be made for including multivariate settings.
> > >
> > > Thank you again for the valuable feedback and for reviewing the paper!

---

### Official Review · Reviewer_5XUS · 2025-03-24

**Overall Recommendation:** 3

**Summary:**

The paper introduces Conformal Relational Prediction (COREL), a novel approach for uncertainty quantification in correlated time series forecasting using graph deep learning frameworks. COREL overcomes the data exchangeability limitation by employing a spatiotemporal graph neural network (STGNN) to model relationships among multiple time series based on past residuals, enabling the estimation of the quantile function for prediction errors. Unlike existing methods that treat time series independently, COREL captures dependencies among correlated series by conditioning uncertainty estimates on neighboring time series through a learned graph structure. Additionally, it incorporates an adaptive mechanism to handle non-stationary inputs, enhancing prediction intervals (PIs) across varying conditions. Empirical evaluations show that COREL achieves state-of-the-art performance across multiple benchmarks, effectively quantifying uncertainty while preserving high predictive accuracy.

**Claims And Evidence:**

The claims in the submission are well-supported by clear and compelling evidence, particularly regarding the performance of the Conformal Relational Prediction (COREL) method. The authors provide empirical results demonstrating that COREL outperforms existing conformal prediction approaches across multiple benchmarks.

The paper also details the algorithmic framework of COREL, highlighting its use of a spatiotemporal graph neural network to capture relationships among time series. Additionally, the inclusion of an adaptive mechanism for handling non-stationary inputs is emphasized as a key contribution, with validation through experimental results.

However, further detailed comparisons with a larger set of existing methods and additional concrete examples of scenarios where COREL outperforms other methods can bolster the robustness.

**Essential References Not Discussed:**

None that I can think of now.

**Experimental Designs Or Analyses:**

1. The paper presents sound experimental designs to validate the effectiveness of COREL, notably through comparative experiments against several baseline methods such as SCP, SeqCP, and SCPI across diverse datasets. This comparative approach is valid but could be strengthened by including a wider variety of baseline models and clarifying their selection rationale.
2. The use of the Winkler score and coverage metrics is appropriate.
3. The controlled environment testing using a graph diffusion process adds rigor but assumes the simulation accurately represents real-world scenarios, which may not always hold. The adaptability experiments evaluating COREL's performance in non-stationary settings are crucial, but more detail on the fine-tuning methods used would enhance transparency.

**Methods And Evaluation Criteria:**

Yes, it generally does.

**Other Comments Or Suggestions:**

No alarming typos.

**Other Strengths And Weaknesses:**

Strengths:

1. COREL introduces a new conformal prediction method that effectively utilizes graph deep learning to quantify uncertainty in correlated time series.
2. The model captures relationships among time series through a graph structure, enhancing forecasting accuracy by leveraging spatiotemporal dependencies.
3. Empirical results demonstrate that COREL achieves state-of-the-art performance compared to existing CP approaches across multiple datasets and scenarios.
4. COREL can be applied to residuals from any point forecasting model, even those that do not consider relationships among input time series, allowing for broader applicability.
5. The inclusion of node-level parameters that adapt to changes in target sequences addresses non-stationarities effectively.

Weaknesses:
1. The integration of graph deep learning may increase computational complexity and resource requirements, potentially limiting its practical deployment in resource-constrained environments.
2. The paper could provide more clarity on the assumptions made about the structure and nature of the underlying time series data, which may affect the generalizability of the findings.
3. While adaptability is addressed, the actual performance in highly volatile or non-stationary conditions is not thoroughly evaluated and discussed.

**Questions For Authors:**

Authors can possibly further explore how well their method performs in broader non-stationary conditions.

**Relation To Broader Scientific Literature:**

The key contributions of the paper, particularly the introduction of COREL for uncertainty quantification in correlated time series using graph deep learning, are closely related to the existing literature in several ways. Firstly, the application of GDL to CP for time seriestraditional.

Conformal prediction (CP) methods, which often assume exchangeability, have been extended to account for time series data, showing limitations in their application due to non-stationarity in real-world scenarios. COREL builds on these findings by integrating a graph structure learning module to capture spatiotemporal dependencies, which has been noted as beneficial for improving prediction accuracy in various applications. Moreover, COREL's reliance on quantile regression aligns with established statistical frameworks for probabilistic forecasting, while addressing the shortcomings of existing CP techniques that often operate independently on univariate time series. By allowing for collective learning across multiple related time series, COREL advances the state of the art in uncertainty quantification methods, demonstrating effectiveness over traditional approaches and laying the groundwork for future explorations in spatiotemporal CP frameworks.

**Theoretical Claims:**

The proof for Prop. 3.1 looks good without any obvious issues.

---

> ### Author Rebuttal · Authors · 2025-03-31
>
> Thanks for the review please find our comments below.
>
> > Further comparisons with existing methods and additional concrete examples of scenarios where COREL outperforms other methods can bolster the robustness. [...] This comparative approach is valid but could be strengthened by including a wider variety of baseline models and clarifying their selection rationale.
>
> Thank you for the comment. The included baselines are representative of the current state of the art and existing approaches to the problem; SCPI and HopCPT, in particular, are modern and very competitive baselines. The paper includes results from 3 datasets and 3 different base models together with 1 synthetic dataset with 2 base models for a total of 11 different benchmark scenarios. We will also include the compute time of the different baselines (see our rebuttal to Rev oUbs). We believe this is an adequate evaluation setup as it highlights the strengths and weaknesses of the different methods in different settings. Nevertheless, we agree that the motivations for choosing the selected baselines could have been discussed more. We will include these considerations in the paper.
>
> > Controlled environment adds rigor but assumes the simulation accurately represents real-world scenarios, which may not always hold.
>
> There might be a misunderstanding. The objective of the experiment on the synthetic dataset is not to mimic a real-world scenario but to show that CoRel can learn and exploit latent relationships among time series. Indeed, its performance here matches that achievable by a model with access to the ground truth relational structure, and its UQ performance is close to the theoretical optimum. For further discussion and analysis, see also our comments on the GPVAR experiment in response to Rev oUbs.
>
> > The adaptability experiments evaluating COREL's performance in non-stationary settings are crucial, but more detail on the fine-tuning methods used would enhance transparency.
>
> Details of the fine-tuning procedure are provided in the appendix. In short, embeddings are updated every M time steps by running the training procedure on the latest observations and keeping all the parameters frozen except for the embeddings. Are there specific aspects that you believe would benefit from more discussion? We will move part of the description of the adaptation procedure to the main body of the paper and we are available for further clarifications.
>
> > W1 The integration of graph deep learning may increase computational complexity and resource requirements, potentially limiting its practical deployment in resource-constrained environments.
>
> The additional computational complexity is inherent in operating on multiple TS simultaneously. Thus, it is more a challenge of the problem settings rather than an issue with our approach. Furthermore, CoRel is much more scalable than the baselines in these settings as discussed in the rebuttal to Reviewer oUbs’ comments. There are also techniques to make the graph-based processing scalable that could be applied here as well. These techniques range from graph sub-sampling (e.g., [1,2]) to scalable architectures (e.g., [3,4]). We will include a discussion of this aspect in the paper.
>
> > W2 The paper could provide more clarity on the assumptions made about the structure and nature of the underlying data, which may affect generalizability.
>
> We do not rely on strong assumptions about the correlated TS besides the existence of Granger causality among them for the graph-based approach to be beneficial. Stationarity assumptions are also discussed in the paper. The sparsity assumption is more of an inductive bias/regularization for learning a sparse graph. If the sparsity assumption is deemed to be unrealistic for a problem at hand, it would be possible to use operators to learn a dense graph, e.g., by modeling each edge as a Bernoulli RV. We will add a comment on this aspect in the paper.
>
> > W3/Q1 While adaptability is addressed [...] authors can possibly further explore how well their method performs in broader non-stationary conditions.
>
> We provide a simple and effective method to make our approach adaptive and show its effectiveness in 3 different scenarios. However, adaptability is not the central focus of our work, which focuses on CP on related TS. We believe that a dedicated study would be needed to address non-stationary settings in depth, as we discuss in the future work section. We will emphasize the importance of this direction in the updated paper.
>
> [1] Hamilton et al. “Inductive representation learning on large graphs” NeurIPS 2017\
> [2] Chiang et al. “Cluster-GCN: An Efficient Algorithm for Training Deep and Large Graph Convolutional Networks” KDD 2019\
> [3] Frasca et al. “SIGN: Scalable Inception Graph Neural Networks” arXiv 2020\
> [4] Cini et al. “Scalable Spatiotemporal Graph Neural Networks” AAAI 2023

---

### Decision · Program_Chairs · 2025-05-01

**Decision:**

Accept (poster)

**Comment:**

This paper considers the problem of uncertainty quantification in relational/spatial time series prediction problems. The proposed approach synergistically combines the benefits of graph deep learning (spatio-temporal GNN) and conformal prediction based distribution-free uncertainty quantification to solve this problem. Specifically, spatio-temporal GNN provides structural embeddings and quantile regression is applied to produce prediction intervals. Experimental results are strong.

This paper received mostly positive reviews, but reviewers' also raised a number of constructive questions/concerns. The author response addresses these concerns and all the reviewers' supported accepting the paper. Therefore, I recommend accepting this paper.